# Development of ISB 1442, a CD38 and CD47 bispecific biparatopic antibody innate cell modulator for the treatment of multiple myeloma

C. Grandclément [1,7], C. Estoppey[1,7], E. Dheilly[1,7], M. Panagopoulou[1], T. Monney [1], C. Dreyfus[1], J. Loyau[1], V. Labanca[1], A. Drake[1], S. De Angelis[1], A. Rubod[1], J. Frei[1], L. N. Caro[1], S. Blein[1], E. Martini[1], M. Chimen[1], T. Matthes [2], Z. Kaya[3], C. M. Edwards [3], J. R. Edwards [3], E. Menoret[4], C. Kervoelen[4], C. Pellat-Deceunynck [4,5], P. Moreau[4,5,6], M. L. Mbow[1], A. Srivastava[1], M. R. Dyson [1], E. A. Zhukovsky[1], M. Perro [1,7] ✉ & S. Sammicheli [1,7] ✉

Antibody engineering can tailor the design and activities of therapeutic antibodies for better efficiency or other advantageous clinical properties. Here we report the development of ISB 1442, a fully human bispecific antibody designed to re-establish synthetic immunity in CD38+ hematological malignancies. ISB 1442 consists of two anti-CD38 arms targeting two distinct epitopes that preferentially drive binding to tumor cells and enable avidity-induced blocking of proximal CD47 receptors on the same cell while preventing on-target off-tumor binding on healthy cells. The Fc portion of ISB 1442 is engineered to enhance complement dependent cytotoxicity, antibody dependent cell cytotoxicity and antibody dependent cell phagocytosis. ISB 1442 thus represents a CD47-BsAb combining biparatopic targeting of a tumor associated antigen with engineered enhancement of antibody effector function to overcome potential resistance mechanisms that hamper treatment of myeloma with monospecific anti-CD38 antibodies. ISB 1442 is currently in a Phase I clinical trial in relapsed refractory multiple myeloma.

Engineering of antibodies is emerging as a revolution in immuno-oncology, allowing for the generation of synthetic immunity by optimizing each component of an antibody structure. Monoclonal antibodies (mAbs) with enhanced antibody-effector functions and bispecific antibodies (BsAb) have been approved for several tumor indications[1–3]. However, tumors evolve quickly and often acquire immune escape mechanisms that may include (a) inhibition of tumor

phagocytosis via upregulation of CD47 'don't eat me' signal, (b) interference with the activities of the complement or NK cells, or (c) clonal escape through antigen downregulation[4–7]. Therefore, novel antibody designs, to overcome these tumor escape mechanisms, are required as next generation cancer immune therapies.

CD47 is a surface protein expressed ubiquitously and is the ligand for the signal regulatory protein alpha (SIRPα), which is constitutively

[1]Ichnos Glenmark Innovation, Lausanne, CH, Switzerland. [2]Haematology Service, Department of Oncology and Clinical Pathology Service, Department of Diagnostics, University Hospital Geneva, 1211 Geneva, Switzerland. [3]Nuffield Department of Orthopaedics, Rheumatology and Musculoskeletal Sciences, Botnar Institute, University of Oxford, Oxford, UK. [4]Nantes Université, Inserm, CNRS, Université d'Angers, CRCI2NA Nantes, France. [5]SIRIC ILIAD, Angers, Nantes, France. [6]Service d'Hématologie Clinique, Unité d'Investigation Clinique, CHU, Nantes, France. [7]These authors contributed equally: C. Grandclément, C. Estoppey, E. Dheilly, M. Perro, S. Sammicheli. ✉e-mail: mario.perro@iginnovate.com; stefano.sammicheli@iginnovate.com

expressed on myeloid cells[8]. The binding of CD47 to SIRPα triggers a signal transduction cascade that leads to the inhibition of phagocytosis[8]. CD47 expression on tumor cells allows them to overcome intrinsic prophagocytic signals, and thereby escape phagocytosis. Blocking of CD47 has demonstrated anti-tumor efficacy both preclinically and clinically[9]. However, because of the ubiquitous expression of CD47 and its high expression on newly formed red blood cells (RBC)[10], targeting CD47 should be carefully designed to prevent both poor pharmacokinetics, due to target mediated drug disposition, and on-target off-tumor depletion of RBCs. Increasing the selectivity to the tumor associated antigen (TAA) and avoiding direct targeting of CD47 has proven effective preclinically[11]. Here we report the development of a multispecific antibody optimized to efficiently overcome the multiple tumor escape mechanisms reported in hematological malignancies due to relapse from anti-CD38 mAb therapies[6], while avoiding non-TAA directed CD47 targeting.

CD38 is a surface protein that is expressed on plasma cells and multiple myeloma (MM) cells[12]. Its specificity has led to the development of daratumumab and isatuximab, two CD38 mAbs now approved for the treatment of MM[13,14]. Targeting CD38 has demonstrated clinical activity in MM and other hematological malignancies expressing CD38, including acute myeloid leukemia (AML) and diffuse large B cell lymphoma (DLBCL)[15]. Anti-CD38 mAbs are initially effective in most MM patients, however the disease inevitably relapses, and the durability of the new available options, such as T cell bispecifics or CAR-T approaches, have yet to be fully demonstrated[16].

Using our proprietary Bispecific Engagement by Antibodies based on the TCR (BEAT®) platform we generated ISB 1442, a CD38xCD47 BsAb for the treatment of MM. ISB 1442 targets MM cells using a biparatopic targeting of CD38 which enables avidity-induced blocking of CD47 receptors on the same cell.

This results in targeting of tumor cells expressing varying levels of CD38 while preventing on-target off-tumor binding of CD47 (e.g. to RBCs). The Fc portion of ISB 1442 is modified to enhance antibody effector functions, including complement dependent cytotoxicity (CDC), antibody dependent cell cytotoxicity (ADCC) and antibody dependent cell phagocytosis (ADCP).

ISB 1442 is currently in a Phase I clinical trial in relapsed refractory multiple myeloma (RRMM).

## Results

### Design and generation of CD38 biparatopic, CD47 bispecific antibody

An avidity-induced approach to selectively inhibit HER3, upon initial binding to tumor cells via HER2, has paved the way to the "dock and block" design of several BsAbs[17,18]. A similar approach demonstrated that a BsAb with a high affinity anti-CD19 can efficiently block CD47 on CD19+ tumor cells employing a low-affinity anti-CD47 Fab domain[19,20]. Further, targeting CD38 with an antibody possessing two non-overlapping epitopes was shown to synergistically increase CDC potency[21]. Therefore, we postulated that an antibody with two high-affinity binding domains targeting distinct CD38 epitopes and a low-affinity anti-CD47 domain could enable efficient avidity-induced blocking of CD47 and enhance Fc functions[20]. ISB 1442 consists of two anti-CD38 Fabs linked through a flexible glycine-serine peptide linker of 15 amino acids (G$_4$Sx3) to the BEAT B chain, while the anti-CD47 Fab is linked to the BEAT A chain (Fig. 1A)[22,23]. To achieve a biparatopic targeting of CD38 independently of the presence of the MM standard of care daratumumab, we selected two Fabs (E2RecA and B6-D9) binding to non-overlapping epitopes on CD38 and not competing with daratumumab. Absence of competition of these Fabs was confirmed by Bio-Layer Interferometry (BLI) (Fig. 1B, C). The non-competitive binding between tested Fabs was quantified by the difference of response units (RU) to immobilized CD38 in the presence or absence of competitor. Despite a slight interference between B6-D9 Fab and daratumumab Fab, possibly due to their binding mode and close epitopes, both Fabs can bind simultaneously to CD38 (Fig. 1B, C). The putative binding sites for the two binders on CD38, in comparison to daratumumab and isatuximab, as measured by epitope binning (Fig. 1 B, C and Supplementary Fig. 2), are shown as ellipses on the surface of CD38 (Fig. 1B). E2RecA and B6-D9 Fabs displayed a $K_D$ to CD38 of 0.9 and 0.55 nM respectively and an apparent $K_D$ of 0.16 nM for CD38 in the 2 + 1 bispecific format (Supplementary Fig. 1 A-C and Supplementary Table 1) of ISB 1442.

To enable CD38-driven avidity binding to proximal CD47 receptors, we selected the H2 Fab, with a CD47 $K_D$ of 0.9 μM in the 2 + 1 bispecific format of ISB 1442 antibody (Supplementary Fig. 1 D, E and Supplementary Table 1), because it was capable of blocking the weak interaction between CD47 and SIRPα ($K_D$ ~ 1 μM) as a monomeric Fab (Fig. 1 D). In 2 + 1 bispecific antibody format, the H2 Fab induced similar CD47-SIRPα inhibition to the high-affinity anti-CD47 (magrolimab, hu5F9) mAb in a CD38+/CD47+ cell-based assay (Fig. 1E). All ISB 1442 binding domains displayed similar binding characteristics for the cynomolgus monkey CD38 and CD47 antigens (Supplementary Fig. 1C, F and Supplementary Table 1). ISB 1442 induced prominent phagocytosis of tumor cells, suggesting that CD47 inhibition was sufficient to enable macrophages to engulf multiple tumor cells in vitro (Supplementary Movies 1 and 2 and Supplementary Fig. 3). The avidity-induced tethering to CD47 did not impact the specificity of ISB 1442, since ISB 1442 selectively bound to Raji cells expressing CD38 (CD38 wild type [WT] or CD38+-CD47-knock out [KO]) while it lacked binding to CD38-KO or CD38-CD47 double KO cells (Fig. 1F). ISB 1442 did not compete with daratumumab for binding to CD38 after saturation of CD38 surface with daratumumab, as measured by BLI while a partial competition was observed when CD38 was saturated with ISB 1442 (Supplementary Fig. 4A). Also, we found 20% inhibition of ISB 1442 binding to CD38 by pre-incubation of CD38+ CD47-KO cells with daratumumab suggesting a potentially low interference of daratumumab with ISB 1442 binding to CD38 antigen on cells (Supplementary Fig. 4B). Of note, this did not affect ISB 1442 potency in vitro (Supplementary Fig. 8).

### Optimization of ISB 1442 architecture and Fc functions

Phagocytes require an activating signal to fully induce phagocytosis upon blockade of CD47-SIRPα axis[24,25]. Further, low ADCC and upregulation of complement inhibitory receptors constitute part of the tumor escape mechanisms by MM cells in patients treated with anti-CD38 mAbs[6]. To fully leverage all Fc effector mechanisms we introduced in the BEAT chain A the S239D and I332E mutations, reported to enhance ADCC and ADCP[26–28] and S324N mutation in both the BEAT chains A and B, shown to enhance CDC[29] (Fig. 1A). Consequently, binding of ISB 1442 to Fcγ receptors was increased 7.0–10.9-fold over trastuzumab containing WT IgG1 Fc domain and 2.9–4.4-fold over ISB 1442 with WT BEAT® Fc, while maintaining invariant the affinity for the human FcRn receptor (Supplementary Table 2). Next, we compared phagocytosis afforded by Fc enhanced, WT and Fc silenced antibodies. The ISB 1442 variant with a silenced Fc did not show measurable binding to Fcγ receptors by SPR (Supplementary Table 2) and did not induce any detectable phagocytosis of tumor cells (Fig. 2A, B). ISB 1442 with the WT Fc induced phagocytosis of target cells, but its activity was lower compared to ISB 1442 with the enhanced Fc (Fig. 2A, B). In addition to ADCP, Fc enhancement in ISB 1442 increased CDC reaching approximately 60% of maximal killing compared to 40% by its WT variant (Fig. 2C, D). Based on these results, ISB 1442 with the enhanced Fc was selected for further development. No detectable CDC were observed by the ISB 1442 variant with a silenced Fc (Fig. 2C, D).

By selectively replacing each Fab by a dummy binder, we assessed the contribution of each Fab domain to the ADCP and the CDC

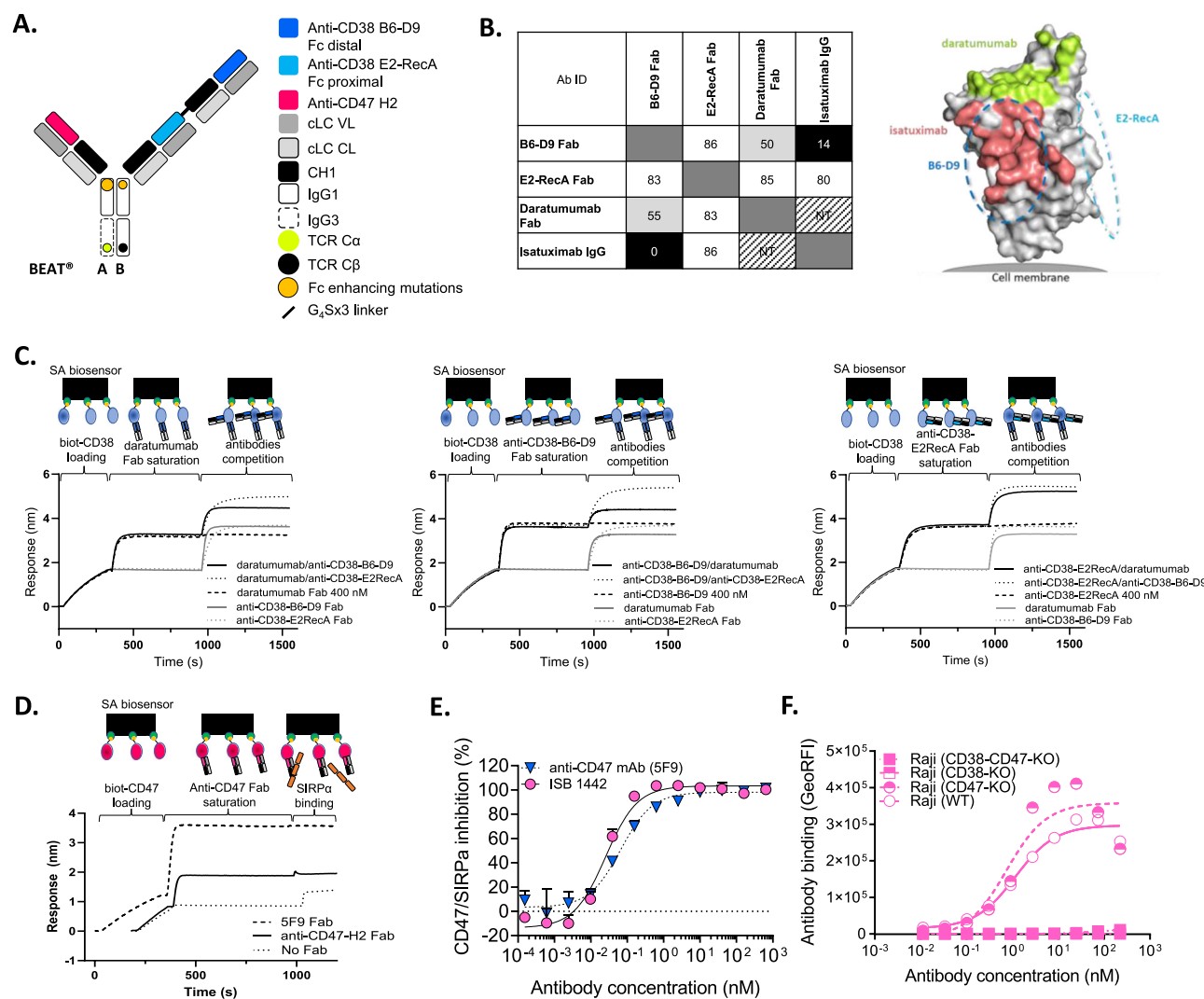

**Fig. 1 | Generation of ISB 1442, a biparatopic CD38 x CD47 bispecific antibody based on the BEAT® platform. A** Schematic view of ISB 1442. Anti-CD38 variable heavy chains (VH) are shown in two different shades of blue. VH of anti-CD47 is depicted in red. All Fab domains make use of an identical common light chain (cLC). Variable (VL) and constant (CL) domains are depicted in gray. Chain A encompasses an engineered human IgG1 CH2 domain with an engineered human IgG3 CH3 domain. Chain B has engineered human IgG1 CH2 and CH3 domains. The BEAT® interface proprietary mutations based on the T cell receptor constant domains alpha (TCR Cα) and beta (TCR Cβ) are depicted by the yellow and black dots. Fc enhancing mutations are depicted by the orange dots. CH constant heavy chain. **B** Left panel: epitope binning results showing plot of saturating antibody in rows against competing antibodies in columns. Values indicate relative percentage of binding of competing antibody to CD38 and are shaded in black, light gray or white whether antibodies are competing, partially competing or non-competing, respectively. Self-blocks are outlined by a dark-gray box. NT not tested. Ab ID antibody identification. Right panel: surface representation of CD38 illustrating the hypothetical epitope bins for B6-D9 (blue dash line) and E2RecA (cyan dash line). The epitopes of daratumumab (PDB 7DHA) and isatuximab (PDB 4CMH) are colored as red and green, respectively. **C** Competition binding assay by Bio-Layer Interferometry (BLI) between anti-CD38-B6-D9 Fab, anti-CD38-E2RecA Fab and daratumumab Fab for binding to CD38. Plots show binding to the sensor tip as a wavelength shift (Response) over time. Curves are labeled by antibodies solution in competition phase. SA streptavidin. **D** Competition binding assay by BLI show blocking of the CD47-SIRPα interaction by anti-CD47-H2 Fab or magrolimab (hu5F9) Fab. Curves are labeled by saturating Fab. **E** Inhibition of the SIRPα-CD47 interaction on CD38high (Daudi) cells by increasing concentrations of ISB 1442 in 2 + 1 bispecific format or anti-CD47 mAb (hu5F9, magrolimab). Representative experiment of at least three independent biological repetitions. Dots represent the mean + SD of $n = 2$ technical replicates. **F** Cell-based binding of ISB 1442 in 2:1 bispecific format to Raji cells WT, Raji CD47-KO, Raji CD38-KO and Raji CD47 and CD38 double KO cells. $N = 1$.

(Fig. 2E). ISB 1442 showed more potent phagocytosis as compared to any monospecific dummy control (Fig. 2F, G) and compared to its two CD38-monoparatopic variants (Fig. 2H, I). Antibody with the anti-CD47 Fab in the absence of the anti-CD38 Fab failed to elicit CDC of tumor cells (Fig. 2J, K). An antibody with only one anti-CD38 binding arm elicited low CDC of tumor cells. However, combining two anti-CD38 targeting two distinct epitopes improved CDC killing potency, which is supported by the lower EC50 of CD38_1+2_DU relative to CD38-1_DU and CD38-2_DU (Fig. 2J, K) or when comparing ISB 1442 to its monovalent anti-CD38 variants, CD47_CD38-1_DU and CD47_CD38-2_DU (Fig. 2L, M). No statistical difference in CDC potency was seen between

CD38-1+2_DU and ISB 1442 (Fig. 2J, K), suggesting that, with biparatopic anti-CD38 targeting, the anti-CD47 arm does not further enhance CDC induction. However, comparing the killing induced by the monovalent anti-CD38-only controls relative to those including CD47 binding, there was a partial rescue of CDC activity due to the CD47-avidity induction that was not observed in the biparatopic format (Fig. 2J–M).

These data suggest that targeting the TAA and CD47 employing a 2 + 1 biparatopic antibody, with the Fc-optimized domain, enhances ADCP and CDC relative to both the non-Fc-optimized IgG1 variant or the Fc-optimized 1 + 1 antibody.

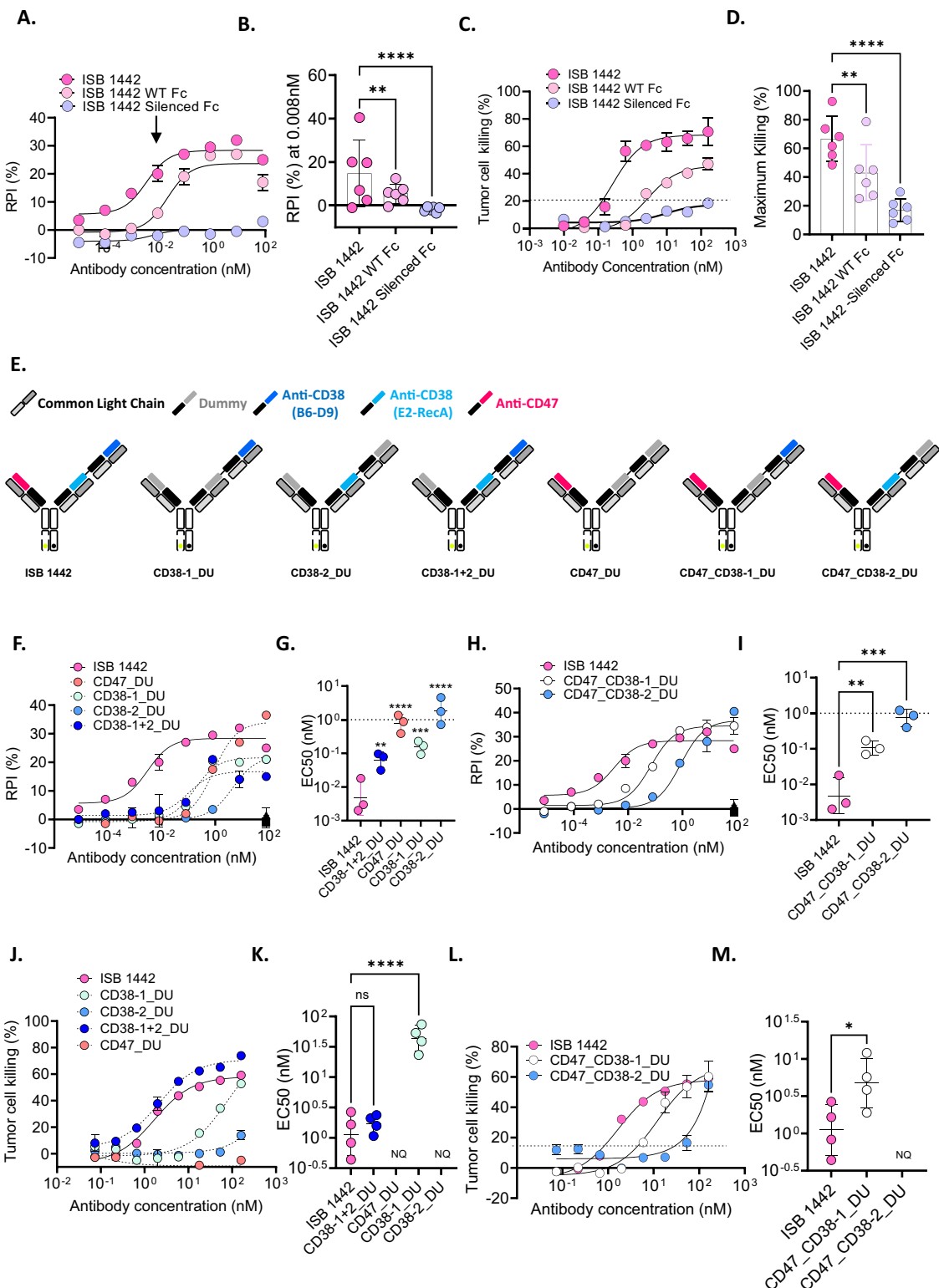

**Fig. 2 | Optimization of ISB 1442 architecture and Fc functions. A** Representative curves of relative phagocytosis index (RPI) of CD38[low] MM cells (KMS-12-BM) induced by ISB 1442 (enhanced Fc) or its respective WT or silenced Fc variants. Dots represent the mean ± SD of $n = 2$ of technical replicates. **B** Cumulative data of RPI as in (**A**). Bars represent mean values of phagocytosis ± SD for 6 donors measured at 0.008 nM (close to the EC50; visualized by an arrow in **A**). **C** Representative curves of CDC on CD38[high] (Daudi) tumor cells. Dots represent the mean ± SD of $n = 2$ of technical replicates. **D** Cumulative data of CDC as in (**C**). Bars represent the mean of maximal killing ± SD in 6 independent experiments. **E** Schematic view of ISB 1442 or its dummy controls. Gray Fabs represent the dummy binders (DU). **F–M** ADCP and

CDC induced by ISB 1442 or its dummy controls. **F–H** Representative curves of RPI on CD38[low] MM cells (KMS-12-BM). Dots represent the mean ± SD of $n = 2$ of technical replicates. **G–I** Cumulative data of EC50 of phagocytosis as in (**F–H**). Bars represent mean values of RPI ± SD for 3 donors. **J–L** Representative curves of CDC on CD38[high] (Daudi) tumor cells. **K–M** Cumulative data of EC50 of phagocytosis of CDC as in **J–L** Bars represent mean values of maximal killing ± SD in 4 independent experiments. Statistics for (**B–D–G–I–K–M**): One-Way Anova with Tukey's multi comparison test, with a single pooled variance. Ns not significant, *$p < 0.05$, ***$p < 0.001$, ****$p < 0.0001$, ****$p < 0.0001$.

## Biparatopic versus monoparatopic CD38 targeting

Antibody density on cell surfaces influences antibody-based effector functions[20,30]. Different thresholds of IgG clustering are required for the induction of ADCC, ADCP and CDC with the latter requiring hexameric assemblies for complement activation[31]. Thus, we postulated that biparatopic anti-CD38 may enable 2:1 antibody:antigen binding, which will enhance CDC compared to monoparatopic 1:1 antibody:antigen binding, similarly to what have been reported previously[21].

To test this hypothesis, we generated monoparatopic variants of ISB 1442, in which both CD38 Fab domains targeted the same epitope (ISB 1442 E2RecAxE2RecA and ISB 1442 B6-D9xB6-D9) and compared their activities to the biparatopic ISB 1442 (Fig. 3A). ISB 1442 showed higher binding to tumor cells as compared to both ISB 1442 E2RecAxE2RecA or ISB 1442 B6-D9xB6-D9 (Fig. 3B, C). The ISB 1442 binding was substantially superior to that of the ISB 1442 E2RecAx-E2RecA variant, however the difference in binding compared to ISB 1442 B6-D9xB6-D9 was less pronounced. ISB 1442 showed a superior killing of tumor cells via CDC as compared to both ISB 1442 E2RecAxE2RecA and ISB 1442 B6-D9xB6-D9 monoparatopic variants (Fig. 3D, E). This suggests that biparatopic targeting enhances CDC of ISB 1442.

Interestingly, we found that cytotoxicity of ISB 1442 was similar to that of the combination of monoparatopic anti-CD38 formats in 1 + 1 or 2 + 1 architectures (Fig. 3F), suggesting that the specific design features of bispecific antibodies can strongly influence their functional activity.

## Potency of ISB 1442 compared to anti-CD38 and anti-CD47 monospecific antibodies

We compared ISB 1442 to clinically validated monospecific antibodies, anti-CD38 (daratumumab) and anti-CD47 (hu5F9, magrolimab) in several potency assays with tumor cell lines expressing varying levels of CD38 and CD47 (Supplementary Table 3). ISB 1442 showed higher binding to CD38$^+$ cells as compared to daratumumab in both high (Daudi and Raji) and low (NCI-H929 and KMS-12-BM) expressing CD38$^+$ cells (Supplementary Fig. 5 and Supplementary Table 4). This is consistent with the biparatopic nature of ISB 1442 as opposed to that of monoparatopic daratumumab, which saturates the CD38 surface receptors at half the amount of ISB 1442 antibody. ISB 1442 showed a comparable level of maximal phagocytosis of CD38$^{high}$ cells as that mediated by daratumumab and hu5F9 (Supplementary Fig. 6A, B). In this cell line, daratumumab showed the highest cytotoxic potency. However, on CD38$^{low}$ cells, ISB 1442 induced an on average 56% phagocytosis, which was significantly higher than the 31% attributable to daratumumab (Fig. 4A, B). Although both CD38$^{high}$ and CD38$^{low}$ tumor cell lines expressed high levels of CD47 (Supplementary Table 3), ISB 1442 phagocytosis was similar to that of high affinity bivalent hu5F9 (Fig. 4A, B and Supplementary Fig. 6). In addition, by using live imaging confocal microscopy, we analyzed phagocytosis of CD38$^+$ and CD38-KO MM cells. While ISB 1442 induced a selective phagocytosis of only CD38$^+$ MM cells, hu5F9 induced similar phagocytosis of both CD38$^+$ and CD38-KO tumor cells (Supplementary movies 3 and 4 and Supplementary Fig. 7A, B), supporting the selectivity of ISB 1442 towards CD38-expressing tumor cells. In addition to phagocytosis, ISB 1442 showed higher tumor cell killing by CDC compared to daratumumab (Fig. 4C, D) and higher potency by ADCC with a comparable maximal killing (Fig. 4E, F).

As physiologically antibody-dependent effector functions occur concomitantly, we established an in vitro multiple mode of action of killing (MMoAK) assay. This assay is based on co-culturing of human PBMCs with tumor cells in the presence of human serum and autologous monocyte-derived macrophages. Test antibodies in this assay can kill tumor cells concomitantly through ADCC (by NK cells), CDC (by complement from human serum), and ADCP (mediated by phagocytes) (Fig. 4G). Importantly, in MMoAK assay, human serum also acts as a source of human immunoglobulins (Igs), allowing assessment of

antibody-dependent effector functions in the presence of physiological levels of Igs, which pre-saturate Fc receptors. Despite the presence of macrophages, hu5F9 induced low level of tumor cells killing in MMoAK, likely due to the competition with serum Igs and the inability to provide activating signaling on macrophages (Fig. 4H, I). In marked contrast, ISB 1442 exhibited prominent tumor cell killing (52%), which was higher than that induced by daratumumab (30%) (Fig. 4H, I). ISB 1442 with its low affinity anti-CD47 and high affinity CD38 binding showed a significantly higher killing of tumor cells compared to the combination of daratumumab and hu5F9 (Fig. 4J, K), suggesting that the delivery of antibody effector signals at the target-effector interface in one molecule is superior to the combination of high affinity anti-CD38 and anti-CD47 mAbs. Finally, neither concomitant treatment with daratumumab (Supplementary Fig. 8) nor soluble CD38 (sCD38) or CD47 antigen sink affected ISB 1442 cytotoxicity (Supplementary Fig. 9).

## ISB 1442 shows improved tumor growth inhibition (TGI) compared to daratumumab in preclinical mouse models

To assess in vivo activity, we used tumor cell lines expressing CD38$^{high}$ (Raji) and CD38$^{low}$ (KMS-12-BM) to generate xenograft models and compared ISB 1442 to daratumumab. In CD38$^{high}$ model, ISB 1442 dosed at 10 mg/kg weekly achieved significant 96% TGI compared to vehicle, while daratumumab, dosed at 16 mg/kg bi-weekly[32], exhibited 2-fold lower TGI of 47% (Fig. 5A). In CD38$^{low}$ model, daratumumab induced a comparable TGI to the vehicle group while ISB 1442 achieved a two-fold higher TGI (Fig. 5B).

Next, we evaluated SC and IV administration of ISB 1442 in a Raji model. ISB 1442 achieved complete tumor regression at a dose of 3 mg/kg bi-weekly IV in 10/10 animals and in 9/10 animals at 3 mg/kg SC, while daratumumab achieved complete tumor regression only in 1/10 animals (Fig. 5C). A group of animals selected randomly were taken forward for survival observation without further treatment and ISB 1442 showed a superior survival compared to daratumumab (Fig. 5D). Of note, only one daratumumab treated animal reached the end of the 60-day observation period, similar to the vehicle, in contrast to ISB 1442 where all animals reached the end of the follow up period of four weeks regardless of route of administration.

Blood PK samples were taken at day 21 (D21) for assessment of trough plasma concentrations for biweekly dosing. A modestly lower exposure was observed for ISB 1442 dosed subcutaneously (SC) as compared to intravenous (IV) route (Fig. 5E). ISB 1442 exposure by both IV and SC routes (18 and 11 µg/ml, respectively) was significantly lower than that of daratumumab (537 µg/ml); nevertheless, ISB 1442 demonstrated higher preclinical efficacy compared to that of daratumumab, confirming the superior potency of the former based on both dose and exposure. ISB 1442 was evaluated in a single dose PK experiment comparing IV and SC dosing at 1 and 10 mg/kg. SC administration gave modestly lower exposure and a greatly reduced Cmax (Fig. 5F). Trough exposures of ISB 1442 seen in tumor (Fig. 5E) were similar to the expected exposure interpolated from the single dose PK (Fig. 5F) indicating that at the dose and schedule examined, there was no large accumulation of ISB 1442 in plasma. Altogether, ISB 1442 was superior to daratumumab in both CD38$^{high}$ and CD38$^{low}$ in vivo preclinical models and, based on the similar elimination PK profiles, D21 exposures and efficacy outcomes when administered SC and IV, we concluded that subcutaneous dosing is effective for ISB 1442 and offers an attractive reduction in Cmax without compromising efficacy.

## ISB 1442 shows significantly lower binding to RBC and lower hemagglutination than hu5F9 ex vivo

CD47-targeting bivalent mAbs are known to induce hemolytic anemia due to their on-target off-tumor depletion of RBCs[33,34]. We compared ISB 1442 with hu5F9 for their effects on RBCs. While ISB 1442 showed no binding on RBCs, hu5F9 showed prominent binding (Fig. 6A). To

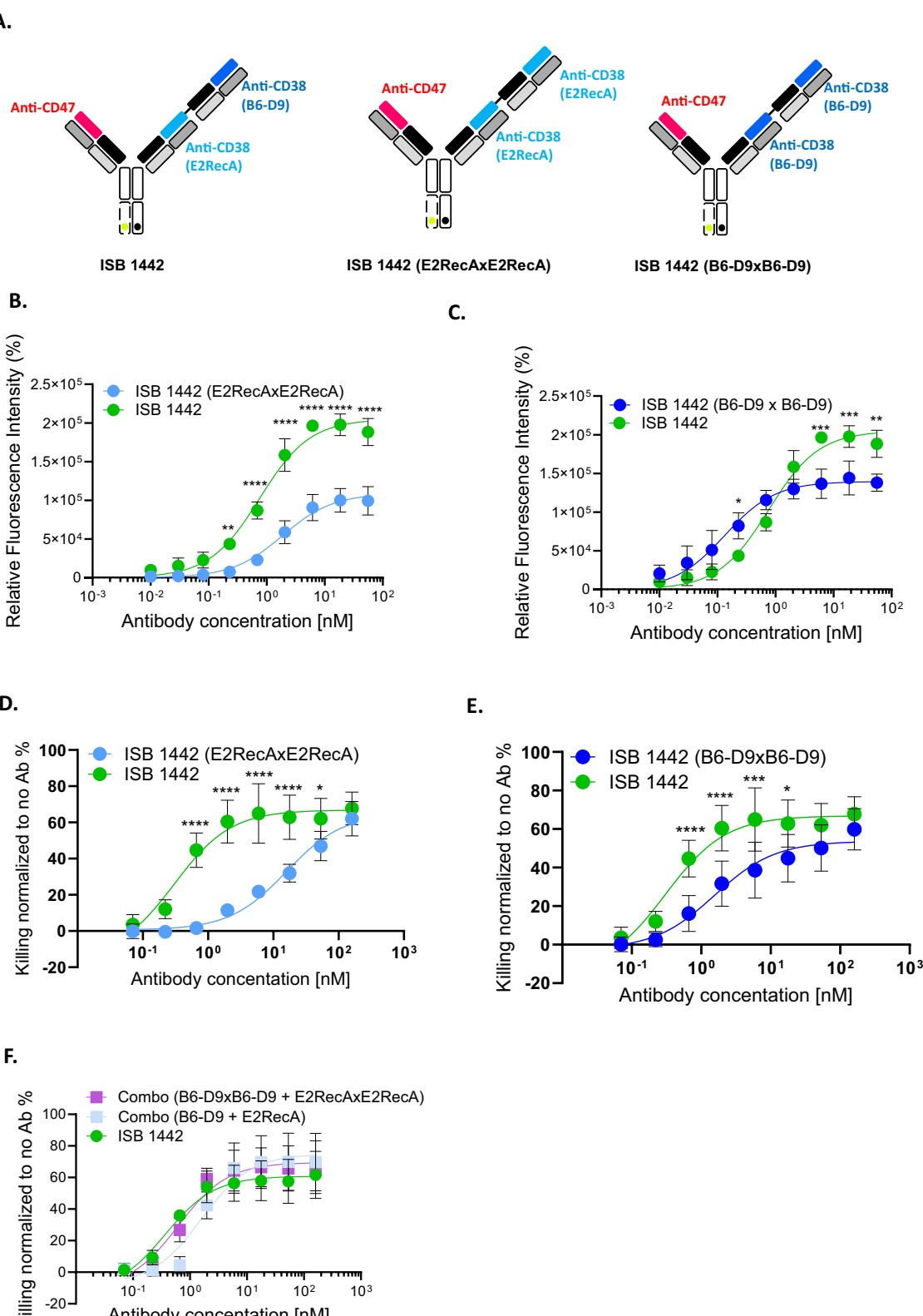

**Fig. 3 | Biparatopic versus monoparatopic CD38 targeting. A** Schematic view of ISB 1442 or 2 + 1 monoparatopic formats. **B**, **C** Representative curves of relative fluorescent intensity of CD38$^{high}$ cells (Daudi) induced by ISB 1442 or its 2 + 1 monoparatopic formats. Dots represent the mean ± SD of $n$ = 3 of biologically independent experiments. **D**, **E** Representative curves of CDC on CD38$^{high}$ (Daudi) tumor cells induced by ISB 1442 or its 2 + 1 monoparatopic formats. Bars represent the mean of maximal killing ± SD in 3 independent experiments. Dots represent the mean ± SD of $n$ = 6 of biologically independent experiments. **F** Representative curves of CDC on CD38$^{high}$ (Daudi) tumor cells induced by ISB 1442 or its 1 + 1 or 2 + 1 monoparatopic formats. Bars represent the mean of maximal killing ± SD in 3 independent experiments. No statistical difference between any condition tested (RM 2-ways Anova). Dots represent the mean ± SD of $n$ = 6 of biologically independent experiments. Statistics of **B**–**E**: *$p$ < 0.05; **$p$ < 0.01; ***$p$ < 0.001; ****$p$ < 0.0001 (RM 2-ways Anova).

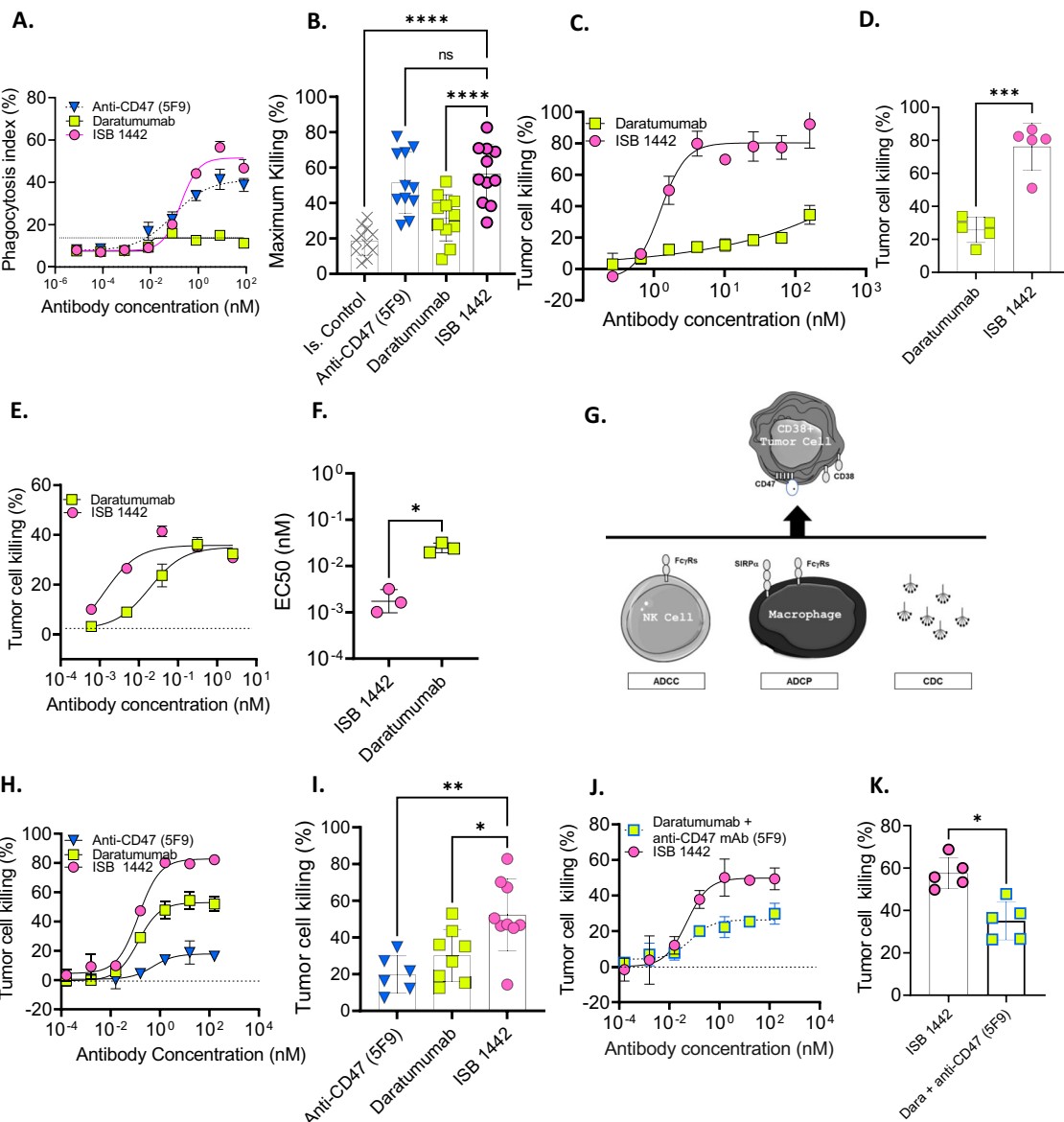

**Fig. 4 | Potency of ISB 1442 compared to anti-CD38 and anti-CD47 monospecific antibody benchmarks. A** Representative curves of phagocytosis index on CD38$^{low}$ multiple myeloma cells (KMS-12-BM). Dots represent the mean + SD of $n = 3$ of technical replicates. **B** Cumulative phagocytosis data as in (**A**). Bars represent mean (± Standard Deviation [SD]) of maximal killing for 11 donors. **C** Representative curves of CDC on CD38$^{high}$ (Daudi) cells. Dots represent the mean + SD of $n = 2$ of technical replicates. **D** Cumulative CDC data as in (**C**). Bars represents mean ± SD of maximal CDC in 5 independent experiments. **E** Representative curves of ADCC on CD38+ multiple myeloma cells (NCI-H929). Dots represent the mean + SD of $n = 2$ of technical replicates. **F** Cumulative EC50 of ADCC data as in (**E**). Bars represent mean of EC50 +/-SD for 3 donors. **G** Cartoon representing MMOAK assay.

**H** Representative curves of MMOAK on CD38+ multiple myeloma cells (NCI-H929). Dots represent the mean + SD of $n = 2$ of technical replicates. **I** Cumulative MMOAK data as in (**H**). Bars represent mean of maximal killing ±SD for 9 donors. **J** Representative MMOAK on CD38+ multiple myeloma cells (NCI-H929) of increasing concentrations of ISB 1442 in comparison to increasing concentrations of daratumumab in combination with anti-CD47 (hu5F9) used at fixed 160 nM. Dots represent the mean + SD of $n = 2$ of technical replicates. **K** Cumulative MMOAK data as in (**J**). Bars represents maximal killing ±SD for 5 donors. Statistics for **B**–**D**–**F**–**I**–**K**: One-Way Anova with Tukey's multi comparison test, with a single pooled variance. Ns: not significant, *$p < 0.05$; **$p < 0.01$; ***$p < 0.001$; ****$p < 0.0001$.

assess the effects of both antibodies on RBCs, we incubated whole blood with ISB 1442 or hu5F9 and counted RBCs using a hematology analyzer. While hu5F9 induced prominent RBCs depletion in whole blood, ISB 1442 did not induce any detectable depletion (Fig. 6B). To detect hemagglutination, we set up an indirect Coombs assay. Hu5F9 induced hemagglutination that had a 33-fold lower EC50 than that induced by ISB 1442 (Fig. 6C, D). We observed neither hemolysis nor platelet aggregation induced by ISB 1442 or any other antibody tested (Supplementary Fig. 10).

Altogether, these data suggest that ISB 1442 demonstrates much lower on-target off-tumor effects to RBCs compared to hu5F9.

## ISB 1442 induced killing of primary MM cells ex vivo
To test ISB 1442 activity under conditions that better reflect clinical disease, we assessed ex vivo the potency of ISB 1442 using bone marrow (BM) aspirates from MM patients. To detect specific tumor cell killing, we developed a flow cytometry-based assay and counted MM cells in primary samples after incubation ex vivo with different concentrations of ISB 1442. ISB 1442 induced a significant increase in killing of MM cells in patient samples compared to isotype controls (Fig. 7A), suggesting that primary tumor cells do not have intrinsic mechanisms protecting them from ISB 1442 killing. ISB 1442 showed higher killing of tumor cells in newly diagnosed MM patients as

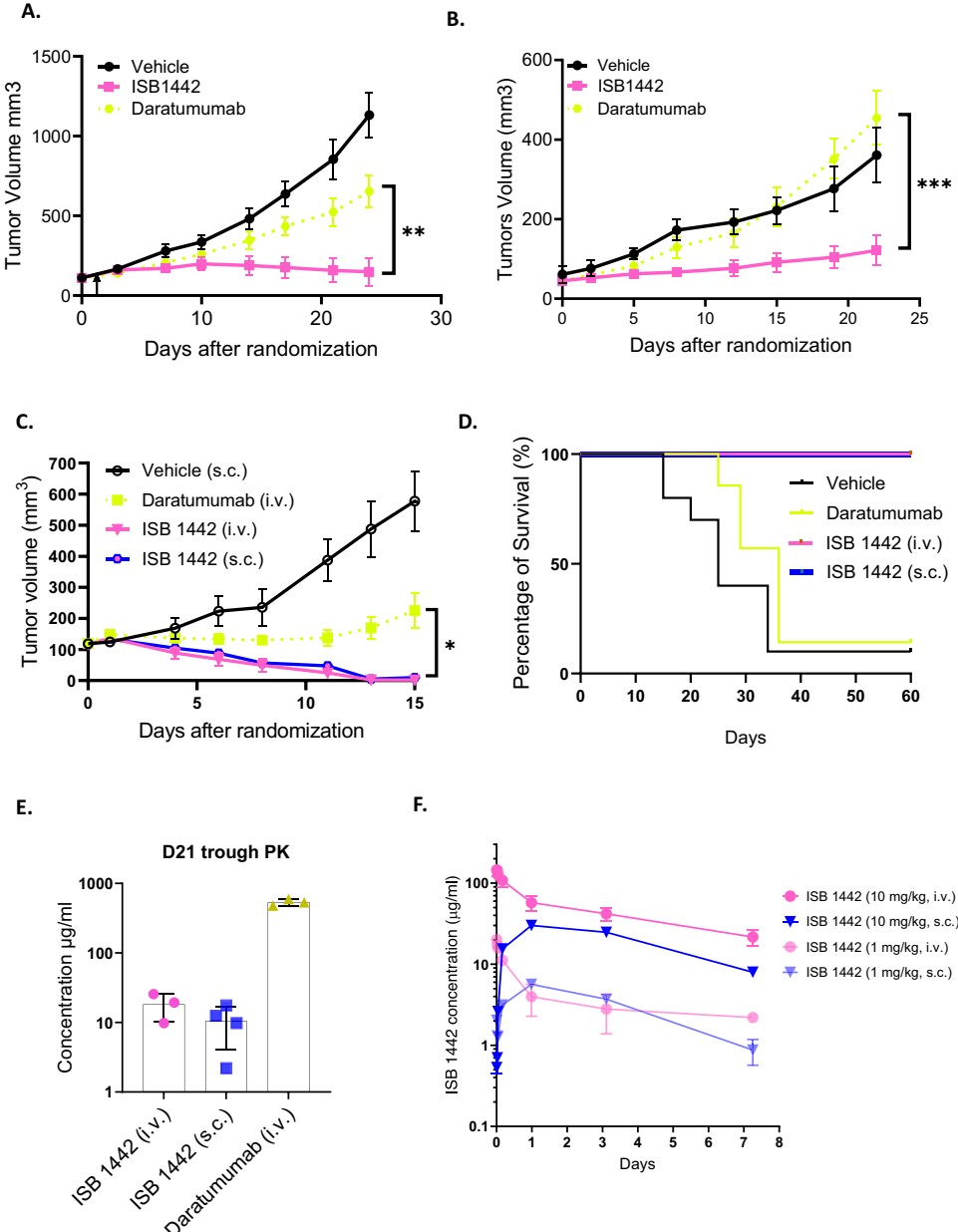

**Fig. 5 | ISB 1442 shows improved tumor growth inhibition compared to daratumumab in preclinical mouse models. A**. CB17/SCID mice (*N* = 5) engrafted with Raji cells (CD38[high]) randomized on D12 and treated intravenously immediately with ISB 1442 at 10 mg/kg weekly, daratumumab at 16 mg/kg twice weekly and PBS (vehicle). **B** CB17/SCID mice (*N* = 5) engrafted with KMS-12-BM cells (CD38 [low]) randomized on D15 and treated immediately with ISB 1442 at 10 mg/kg weekly, daratumumab at 16 mg/kg biweekly and PBS (vehicle). **C** CB17/SCID mice engrafted with Raji cells treated twice weekly with daratumumab at 16 mg/kg intravenously (i.v.), and ISB 1442 at 3 mg/kg both intravenously (i.v.) and subcutaneously (s.c.). **D** Kaplan–Meier survival curve for experiment shown in panel **C** excluding 3 mice taken randomly for terminal analysis from each group at D15 unless they had

reached the experimental endpoint (*N* = 6–9), survival endpoint was deemed to be reached when animals exceeded maximum permissible tumor volume of 1000 mm³. **E** Trough PK samples from 21 days post randomization (3 days post last dose) of study shown in (**C**, **D**). **F** PK time course at 1 and 10 mg/kg doses of ISB 1442 subcutaneously (s.c.) and intravenously (i.v.) in BALB/c nude mice (*N* = 4). Statistical evaluation by 1-way ANOVA followed by Tukeys post hoc test on final day of experiment prior to the humane endpoint for the second mouse in any group (**A**–**C**). Log Rank (Mantel Cox) test used to evaluate significance of difference in (**D**). *P* values for tests presented on (**A**–**C**) **A** *P* = 0.0076, **B** *P* = 0.0009, **C** *P* = 0.016 (IV vs. Dara) 0.0222 (SC vs. Dara) Ns: not significant, \**p* < 0.05; \*\**p* < 0.01; \*\*\**p* < 0.001; \*\*\*\**p* < 0.0001.

compared to daratumumab and a much higher tumor cell killing than that induced by daratumumab of tumor cells from RRMM patients (Fig. 7B, C). We found no significant difference between cytotoxicity of ISB 1442 and daratumumab in RRMM patients that were not exposed to CD38-therapies (Fig. 7D). However, ISB 1442 induced strong killing of tumor cells from patients treated with anti-CD38 therapies, whereas daratumumab induced only a low level of killing (Fig. 7E). Altogether these data suggest that ISB 1442 retained cytotoxicity against tumor cells from patients with daratumumab-refractory MM.

## Discussion

Antibody engineering is unlocking tremendous opportunities for cancer immune therapies by enabling the design of innovative therapeutic agents. Here we described ISB 1442, a fit-for-purpose multispecific antibody, rationally designed to harness innate immunity to treat CD38+ hematologic malignancies. ISB 1442 antibody is distinguished due to three features: (i) it uses two distinct Fab arms to target a tumor associated antigen (biparatopic approach), allowing for improved CDC and improved binding to tumor cells when an antigen is

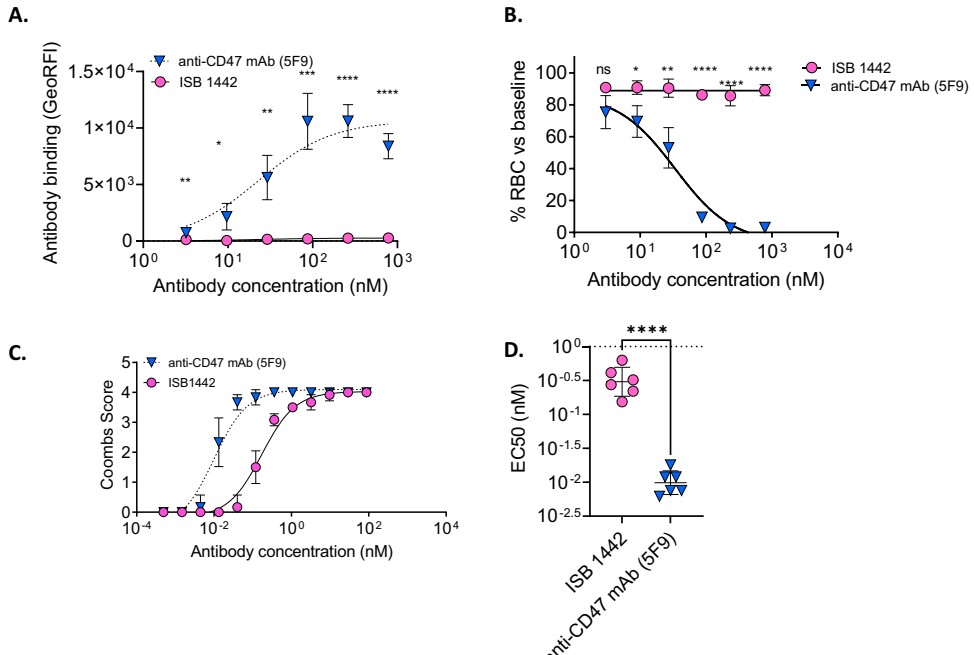

**Fig. 6 | On-target off-tumor on RBCs. A** Means of the binding of ISB 1442 and hu5F9 on human fresh RBCs from 6 independent donors at increasing concentrations. Each dot represents a biological replicate (±SD). Statistics: *$p < 0.05$ **$p < 0.01$ ***$p < 0.001$ ****$p < 0.0001$ (RM 2-ways Anova). **B** RBCs quantification using hematology analyzer. Dose-response of RBCs depletion (in %) as a function of ISB 1442 or hu5F9 concentrations with 6 independent human fresh peripheral blood donors.

Statistics: *$p < 0.05$ **$p < 0.01$ ***$p < 0.001$ ****$p < 0.0001$ (RM 2-ways Anova). **C** Indirect antiglobulin Coombs test. Dose–response of the indirect Coombs score as a function of ISB 1442 or hu5F9 concentrations with 6 independent human fresh peripheral blood donors. **D** Scatter dot plot showing mean ± SD of EC50 of coombs score as in (**C**). Each dot represents a biological replicate. Statistics: One-Way Anova with Tukey's multi comparison test. ****$p < 0.0001$.

downregulated; (ii) it blocks the CD47 'don't eat me' signal to counteract tumor escape from phagocytosis, leveraging selective avidity-induced binding to CD38+ tumor cells, thereby avoiding off-tumor targeting; and (iii) it is equipped with the Fc mutations enhancing effector mechanisms (CDC, ADCC and ADCP). The flexibility of the BEAT® platform, which allows for the generation of multispecific antibodies with drug-like properties, enabled straightforward integration of all these features into a single antibody molecule[35].

Therapeutic targeting of CD47 is challenging due to its ubiquitous expression on healthy cells. Hematotoxicity and poor pharmacokinetics are common unwanted side effects reported in clinical studies[36]. In addition, it has recently been reported that anti-CD47 mAb magrolimab was not superior relative to the standard of care (SOC) in two phase III trials in patients with high-risk myelodysplastic syndrome (HR-MDS) and AML when used in combination with the SOC. As Fc-FcγR interactions are required for anti-tumor activity of anti-CD47 antibodies[37], the lack of superior activity could in part be attributed to lack of the effector function in magrolimab's IgG4 Fc. ISB 1442 overcomes several of these deficiencies: i) it blocks CD47 only on CD38+ cells, and ii) it possesses the enhanced binding to FcγR thus enabling augmented ADCP and the other effector functions. Improving selectivity of CD47 blocking by employing BsAbs is an established approach, which was applied to targeting a diverse set of TAAs: CD19+[19,38], CD20+[39], PDL-1+[40] Mesothelin+, EGFR+ and HER2+[36]. However, all these BsAbs represented the 1 + 1 format, constructed with either IgG4 (silenced effector function) or IgG1 (native effector function) making 2 + 1 biparatopic bispecific ISB 1442 with enanched Fc effector function a unique approach.

To further increase the selectivity towards the TAA, we used a 2 + 1 BsAb format using high affinity, biparatopic anti-CD38 Fab domains. Antibody opsonization required for reaching the Fc-dependent activation threshold is defined by multiple parameters, including antibody antigen affinity, valency and antigen expression density[20]. For T cell

redirecting BsAbs, Bluemel et al. showed that cytotoxic potency can be affected by the targeted epitope and thus by the geometrical arrangement created on the cell surface[41]. Using dedicated dummy Fab domains to replace Fabs of ISB 1442 and by comparing the variants with biparatopic and monovalent anti-CD38 binding, we found that biparatopic antibodies displayed superior binding to tumor cells and induced increased phagocytosis and CDC compared to antibodies targeting a single epitope on CD38, even when the latter possessed Fc enhancing mutations. This suggests that the 2 + 1 design not only improves targeting but also results in superior cytotoxicity compared to the 1 + 1 BsAbs in development. Comparing ISB 1442 to 2 + 1 monoparatopic variants, we found that biparatopic ISB 1442 induced superior CDC relative to that of the variants, in which two anti-CD38 Fab domains targeted the same epitope. The combination of both 2 + 1 monoparatopic variants displayed similar CDC to that of ISB 1442. Therefore, our data suggest that biparatopic bispecific antibodies may replace the need of development of antibody combinations by achieving comparable activity to that of two monoparatopic bispecific antibodies combined, and thus greatly simplify clinical development.

Since Fcγ receptor activation is required for full induction of phagocytosis[24], we found that ISB 1442 variants with a silent Fc, and capable only of blocking the CD47-SIRPα inhibitory axis, did not induce detectable phagocytosis. The ISB 1442 variant with the IgG1 domain induced phagocytosis, however its activity was lower compared to ISB 1442 possessing Fc enhancing mutations. Our rationale to increase affinity to Fcγ receptors and enhance effector function in combination with blocking of CD47 with a low affinity binder, was to achieve the broadest therapeutic window and the highest chance of success in treating patients with ISB 1442 as a monotherapy. Indeed, therapeutic antibodies compete with circulating IgGs for binding to Fcγ receptors when injected into patients. This is of particular importance for MM, a plasma cell proliferative disease characterized by high level of circulating antibodies. In addition, in the tumor

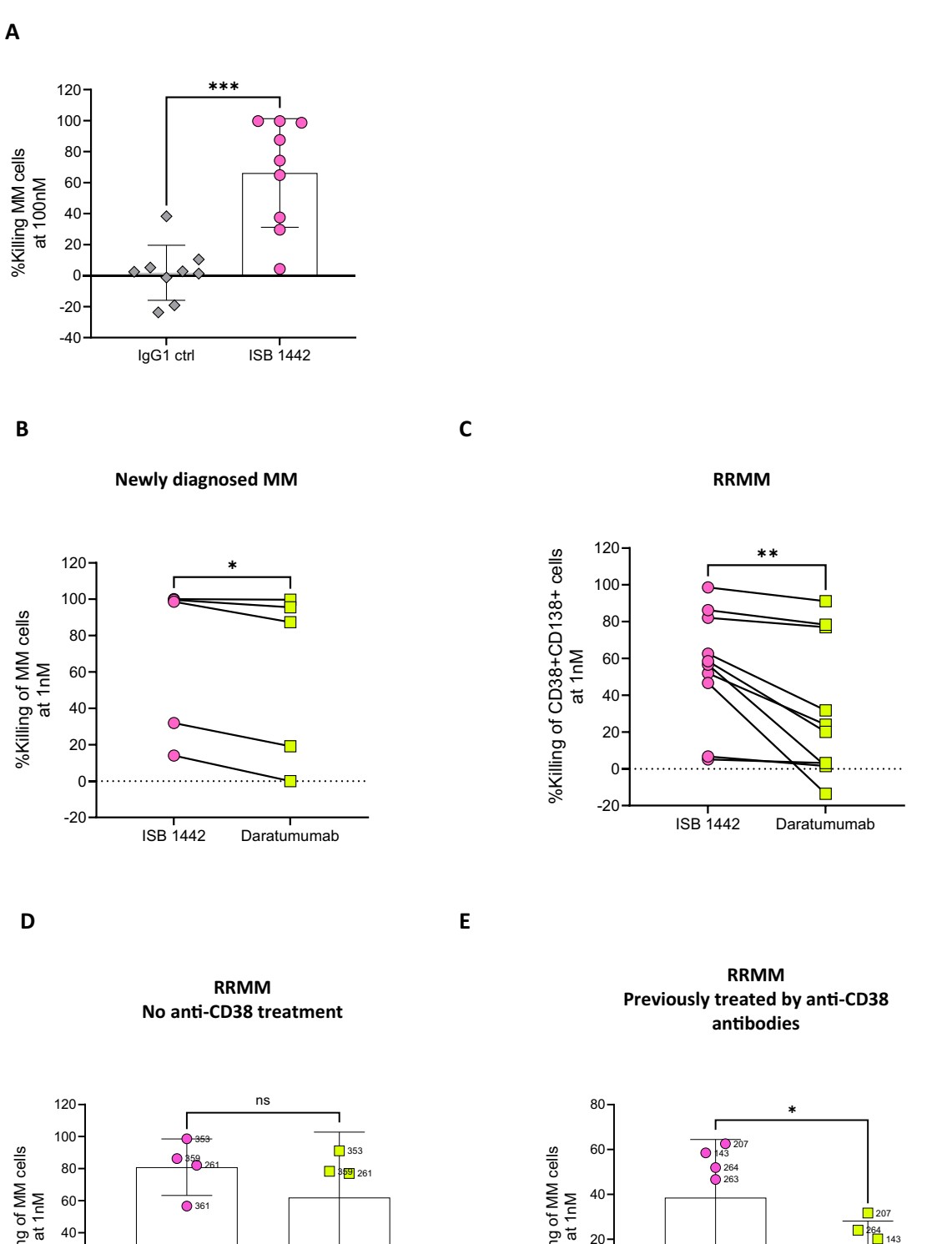

**Fig. 7 | ISB 1442 induced killing of tumor cells in MM patient samples ex vivo.**
**A** Compiled analysis of CD138 + CD38+ cell killing induced by ISB 1442 as compared
to isotype control (IgG1) antibody ($n = 9$ bone-marrow samples). **B**–**C**. Compiled
analysis of CD38 + CD138+ cell killing induced by ISB 1442 at 1 nM in 5 newly
diagnosed patient samples (**B**) and in 10 RRMM patient sample (**C**). **D**, **E** Compiled

analysis of killing induced by ISB 1442 in RRMM samples with no prior anti-CD38
mAb treatments (**D**) or previously exposed to anti-CD38 therapies (**E** Patients 274
and 263 treated with isatuximab, patients 207, 264, 143, 059 treated with dar-
atumumab). Statistics for panels **A**–**E**: two tailed paired *T*-test, *$p < 0.05$;
**$p < 0.01$; ***$p < 0.001$.

microenvironment, multiple immune inhibitory mechanisms are in place, specifically inhibiting antibody effector functions, by upregulation of complement inhibitory receptors, downregulation of FcγRIIIa on NK cells, in addition to CD47 upregulation, which is balancing the Fc-induced phagocytosis upon target cell opsonization[6]. Therefore, enhancing Fcγ receptor binding could be a differentiator for improved engagement of effector cells in tumor tissues compared to all current BsAbs targeting CD47 possessing either IgG1 or IgG4 domains. We show that enhancing the effector function of ISB 1442 did not target the RBCs as was reported for other BsAbs targeting CD47; ISB 1442 did not deplete RBCs, in stark contrast to hu5F9 (magrolimab). Altogether, this is expected to result in a better therapeutic index of ISB 1442 in patients compared to high affinity anti-CD47 antibodies.

Targeting CD38 with daratumumab as a single agent and in combination with proteasome inhibitors or immune modulatory drugs has improved median overall survival (OS) and progression free survival (PFS) of MM patients in multiple trials[13,14,42–44]. Since relapses post-daratumumab therapy occur[6], we generated ISB 1442 with anti-CD38 binding domains targeting two different epitopes on CD38 than that of daratumumab, and allow the treatment of patients without the need for a 6-month washout period[45]. In all in vitro assays tested, including CDC, ADCP, ADCC and the MMoAK, ISB 1442 showed superiority against daratumumab, its potency was not impacted by pre-bound daratumumab, and it was superior to the combination of daratumumab and hu5F9. This implied that the engineering and architecture of ISB 1442 possesses improved effector functions compared to the combination of separate antibodies targeting the same antigens.

ISB 1442 consistently demonstrated prominent tumor growth inhibition compared to daratumamab and led to superior survival in CB17/SCID mice, chosen for their intact Fc mediated immune mechanisms[46]. This outcome was independent of CD38 expression level and was more prominent in the stringent CD38$^{low}$ MM model, in which daratumumab was inactive. This result suggests that ISB 1442 could benefit MM patients relapsing from daratumumab. Indeed, ISB 1442 induced killing of tumor cells in patient samples was superior to that of daratumumab and was significantly higher in samples from patients that relapsed from the anti-CD38 therapy; these observations suggest that ISB 1442 could be used as a salvage therapy.

ISB 1442 may represent an improvement of the monospecific anti-CD38 antibody therapeutics (e.g. daratumumab and isatuximab) in MM. ISB 1442 is the first, and to our knowledge, the only anti-CD47 bispecific antibody equipped with the enhanced Fc effector function; our preclinical data suggest that enhancing the Fc function, combined with low affinity anti-CD47 targeting, is key for superior ISB 1442 activity. These data are in line with the recent report that anti-CD47 antibodies require an Fc contribution for anti-tumor activity[37]. Anti-CD47 magrolimab and other anti-CD47 mAbs have recently been halted in the MDS and AML clinical trials; however, clinical development in solid tumors of magrolimab and other anti-CD47 mAbs continues and this will clarify whether the lack of efficacy of these antibodies was specific only to leukemia.

ISB 1442 is being evaluated clinically (Trial ID: NCT05427812). This dose escalation and expansion trial aims to explore tolerability, efficacy and PK. PK, specifically target mediated drug distribution, is a known challenge for CD38 monotherapies especially at lower doses and early in treatment; since ISB 1442 interacts with CD38, CD47 and Fc target mediated drug distribution is expected to differ from monospecific agents and this will be investigated during the ongoing trial. This trial will also help elucidating specific mechanisms of action associated with three functional domains in ISB 1442, and whether our strategy of increasing its therapeutic index provides substantial clinical benefit relative to other anti-CD38 antibodies (e.g. daratumumab and isatuximab) while overcoming serious toxicities associated with high affinity monospecific anti-CD47 antibodies.

In conclusion, we have described here the application of the BEAT® multispecific antibody technology to generation of ISB 1442, a bispecific biparatopic antibody with the ability to induce synthetic immunity via multiple effector mechanisms, while bypassing several tumor escape mechanisms that attenuate activity of monospecific antibodies[6]. ISB 1442 is envisioned to be a next generation innate immune cell engager for CD38$^+$ hematologic malignancies and is currently in a Phase 1 clinical trial for RRMM.

## Methods

### Animals
All in vivo experiments were performed in female 6-7-week-old immune-deficient CB-17/Icr-Prkdc scid/Rj mice from JANVIER LABS (catalog CB-17/Icr-Prkdc scid/Rj) except PK which was performed in JAX CBySmn.Cg-Prkdcscid/J (catalog #001803). All mice were maintained under standardized environmental conditions in rodent cages. All studies were performed on the ethical approval of either Transcure (Archamps, France) or Ichnos Sciences (Swiss FSVO license, Licenses 33147 and 33137). Tumor cells were engrafted subcutaneously and were dosed either intravenously or subcutaneously (opposite flank to tumor) at the doses described for the specific experiment. For both Raji and KMS-12-BM, $10^7$ cells were engrafted in PBS. Tumor volume was determined by caliper measurement and TGI was calculated as follow: TGI = (1 − (Treatment Vcurrent − Treatment Vstart)/(Control Vcurrent − Control Vstart)) x 100. For ethical reasons (avoidance of single housing post randomization) only female mice were used so no consideration of sex was possible.

Institutional maximum tumor size was 1000 mm³, as ethically agreed all animals were humanely euthanized if their tumor was measured exceeding 1000 mm³. When maximum tumor volume was reached mice were humanely euthanized by $CO_2$ asphyxiation as approved in our animal licenses. Mice were SPF housed for all experiments.

### Human samples
All research on healthy human donor was approved by transfusion Interregionale CRS, with all donors provided written informed consent in accordance with the Declaration of Helsinki and the protocol of the local institutional review board, the Medical Ethics Committee of Transfusion Interregionale CRS. Human PBMCs from healthy donors and bone marrow mononuclear cells (BMMCs) were isolated using Ficoll gradient according to manufacturer instructions. All research on MM patient's samples were performed in accordance with ethical approvals with collaborating institutes with patients providing written informed consent. Oxford Clinical Research: The study was approved by the Oxford Clinical Research Ethics Committee (17/SC/0572) and the HaemBiobank Governance Committee (BBProj-27.0 and BBProj-13.0). University Hospital Geneve (HUG): ethical approval number 2021-02416. Nantes: all samples were obtained from the cohort MYRACLE (NCT03807128)[47]. Gender was not considered in the study design due to limited availability of patients' samples, priority was given to different treatment history.

### Cell culture
Raji cell line was purchased from ATCC. KMS-12-BM, NCI-H929, MOLP8 and Daudi were obtained from DSMZ. Raji-KO and NCI-H929 KO cell lines have been generated in-house using CRISPR/CAS9 technology and clones isolated using Fluorescence-Activated-Cell sorter (FACS). All cell lines were cultured in RPMI 1640 medium (Gibco) supplemented with 10% heat-inactivated fetal bovine serum (FBS,Gibco), 2 mM L-glutamine (Gibco), 1% non-essential amino acids (Gibco) and 1 mM sodium pyruvate (Gibco) and maintained at 37 °C under an atmosphere containing 5% $CO_2$. Mycoplasma and short tandem repeat (STR) analysis was routinely evaluated by Microsynth (Balgach, Switzerland) passage 5 and passage 15 according to Microsynth guidelines. No commonly misidentified cell lines were used (according to ICLAC register version 10).

## Antibodies/Treatment

Daratumumab was purchased from Janssen Biotech Inc. Anti-CD47 monoclonal antibody (hu5F9) was produced in house by transient expression into suspension-adapted HEK293-EBNA cells (catalog no. ATCC-CRL-10852, LGC Standards, Teddington, UK) using poly-ethyleneimine. Briefly, equal quantities of each engineered chains vectors were co-transfected. The culture was fed with the same volume of Ex-cell 293 media and incubated for 5 days at 37 °C, 150 rpm. The supernatant was then harvested by centrifugation at 3220 × g for 15 min and sterile filtered (0.22 μm) prior purification by Protein A and cation exchange chromatography, as described below.

## Transient expression and purification

ISB 1442 and other antibody constructs were expressed transiently in CHO-S cells. Engineered chains vectors and a vector encoding Epstein-Barr Virus (EBV) nuclear antigen-1 (EBNA-1) were co-transfected into CHO-S cells (cGMP banked, Invitrogen), using polyethyleneimine (PEI). Typically, cells were prepared at 8 million cells per ml in CD-CHO media (Gibco). Cells were then transfected with a DNA-PEI mixture at 37 °C. Four hours post-transfection, the cell culture was diluted 1:1 in PowerCHO™ 2 (Lonza) supplemented with 4 mM L-Glutamine and incubated for 14 days with orbital shaking at 32 °C, 5% $CO_2$ and 80% humidity. Clarified cell culture supernatants containing the recombinant proteins were prepared by centrifugation followed by filtration. Purification of antibody constructs from supernatant by protein A was performed as previously described[22]. For hu5F9, the mAb was eluted with glycine 0.1 M pH 3.5. Subsequently, the molecules were optionally taken through a second step of purification by cation exchange chromatography to reach monodispersity >95%, as judged by analytical size exclusion chromatography.

## Affinity measurements to CD38 and CD47 by Surface Plasmon Resonance (SPR)

SPR analysis was used to measure the association and dissociation rate constants for the binding kinetics of ISB 1442 to recombinant human CD38, cynomolgus monkey CD38, human CD47 and cynomolgus monkey CD47 proteins. The binding kinetics were measured at 25 °C on a Biacore 8 K+ instrument (Cytiva). For binding to CD38, ISB 1442 was immobilized on a Series S CM5 Sensor Chip previously coupled with Anti-Human IgG (Fc) antibody (Cytiva) and increasing concentrations of his-tagged human CD38 protein (Acrobiosystems) or his-tagged cynomolgus monkey CD38 protein (Acrobiosystems), from 0.14 nM to 100 nM (1:3 dilution series), were flushed onto the immobilized ligand. For binding to CD47, biotinylated human CD47 or biotinylated cynomolgus monkey CD47 protein was immobilized on a Biotin Capture (CAP) Sensor Chip (Cytiva) and increasing concentrations of ISB 1442, from 2.7 nM to 2 μM (1:3 dilution series) were flushed onto the immobilized ligand. The experimental data were fit using a 1:1 Langmuir binding model on double reference subtracted sensorgrams (reference surface and buffer injection subtractions). Measurements were performed in at least three independent replicates.

## Affinity measurements to Fc receptors by SPR

Binding of ISB 1442, a version of ISB 1442 with a wild-type Fc (without Fc enhancing mutations), a version of ISB 1442 with silenced Fc functions and trastuzumab to his-tagged human FcγRI extracellular domain (ECD) (Acrobiosystems) and his-tagged human FcγRIIa ECD, FcγRIIb ECD, FcγRIIIa ECD and FcRn ECD & β2 microglobulin produced internally was evaluated by SPR on a Biacore 8 K+ instrument (Cytiva) at 25 °C. The antibody constructs were immobilized on a Protein L Series S CM5 Sensor Chip (Cytiva) and increasing concentrations of Fc receptors were flushed onto the immobilized antibody constructs. Data were fit to a 1:1 binding model (FcγRI and FcγRIIIa) or to a steady state affinity model (FcγRIIa, FcγRIIb and FcRn) on double reference

subtracted sensorgrams (reference surface and buffer injection subtractions). Measurements to FcγRI, FcγRIIa, FcγRIIb and FcγRIIIa were performed at neutral pH, while measurements to FcRn were done at pH 6.0. Measurements were performed in at least three independent replicates.

## Competition assays by Bio-Layer Interferometry (BLI)

Competition of antibodies was assessed using BLI. Measurements were done on an OctetRED96e instrument (Sartorius) at 25 °C and with shaking of the plate. Streptavidin biosensor (Sartorius) coated with biotinylated human CD38 antigen (Acrobiosystems) were dipped into a solution of 200 nM of a saturating antibody for 10 min to reach saturation of the CD38 coated surface, followed by a successive dip into a mixed solution of 200 nM of the same and 200 nM of a competing antibody. Saturation of the CD38 sensor surface was verified by dipping antibody-saturated CD38 sensor surface into a solution of the same antibody at 400 nM. Controls including competing antibody alone were included to assess maximum binding of competing antibody in absence of saturating antibody. Data was analyzed using Octet HT 11.1 software and user-defined thresholds on percentage of binding of competing antibody to CD38 relative to maximum binding in the absence of saturating antibody were applied to classify antibodies as either competing (<30%), partially competing (>30%<70%), or non-competing (>70%) pairs.

## Absolute quantification of surface antibodies bound to cells (sABC)

Quantification of CD38 and CD47 on cell surface were determined using Biocytex® kit according to the manufacturer's instructions.

## Flow cytometry assays

Tumor cell lines or fresh PBMCs were incubated with increasing concentrations of antibodies or an isotype control at 4 °C for 30 min. Cells were washed twice followed by an incubation with a PE or APC-fluorescently labeled monoclonal anti-human IgG secondary antibody (Biolegend) and incubated at 4 °C for 30 min. Cells were washed twice and resuspended in FACS Buffer (PBS 2.5% FCS, 2 mM EDTA and 0.05% NaN3) containing a viability dye (Dapi or Sytox). Samples were acquired on a Cytoflex cytometer (Beckman Coulter). Data were analyzed using FlowJo software (BD). PE or APC Geometric Mean of Fluorescence Intensities (geoMFI) of viable single cells for each sample were extracted. A normalization was then performed using the geoMFI of each isotype control antibody. The values of geoMFI from the control staining were subtracted to the geoMFI values of each molecule to generate the relative fluorescence intensity (RFI).

## SIRPα/CD47 BLI blocking assay

Measurements were done on an OctetRED96e instrument (Sartorius) at 25 °C and with plate shaking. Streptavidin SA Biosensor (Sartorius) coated with biotinylated human CD47 protein was dipped into a solution of 1 μM of a saturating Fab, followed by a successive dip into a mixed solution of 1 μM of the same Fab and 2 μM of recombinant human SIRPα protein.

## SIRPα/CD47 cell-based blocking assay

Daudi cells labeled with Hoescht (Thermofisher) were incubated with increasing concentration of test antibodies or control antibodies for 30 minutes at 4 °C. In parallel, a mix of antibodies (detection mix) composed of 300 ng/mL SIRPα-mIgG1 Fc fusion protein (Acro Biosystem) and 2.5ug/mL of Alexa Fluor 647-conjugated anti-mouse IgG Fc detection reagent (Jackson Immunoresearch) was prepared in FACS buffer and incubated for 30 min at room temperature in the dark. The detection mix was added to the cells and incubated for 3 h at room temperature. Binding of recombinant SIRPα-Fc to tumor cells was

quantified by image-based analysis using CellInsight CX5 High Content Screening Platform (Thermofisher). CD47/SIRPα inhibition is shown as percentage of inhibition of the Detection Mix fluorescence signal.

## Antibody-dependent-cellular phagocytosis assay

CD14$^+$ monocytes were isolated from fresh human PBMCs using EasySep Monocytes Isolation kit (StemCell) according to the manufacturer's instructions and differentiated into macrophages using M-CSF (50 ng/ml) for 7 days. On the day of the assay, pHrodo-labeled tumor cells were incubated with Cell-Trace Violet-labeled monocytes-derived-macrophages for 1 and half hour at 37 °C, 5% $CO_2$ in presence of increasing doses of test antibodies. Phagocytosis was evaluated using CellInsight CX5 High Content Screening Platform (Thermofisher) and analyzed using HCS Studio Cell Analysis Software (Thermofisher). Phagocytosis was quantified using image-based analysis as the average number of pHrodo-bright tumor cells for 100 cell trace violet-positive macrophages. Relative Phagocytosis Index (RPI) was calculated according to the following formula: RPI=Phagocytosis Index (tested antibody) - Phagocytosis Index (baseline-untreated condition).

## Complement-Dependent-Cytotoxicity Assay

Tumor cells were labeled with 5 μM calcein AM and plated in presence of 50% human serum, for 4h30 at 37 °C, 5% CO2. Triton X-100 was used as a positive control for maximum tumor cell killing. After the completion of the assay, cells were centrifugated and fluorescence induced by calcein release was determined using a Synergy Plate reader at 485/515 nm. Specific killing was calculated according to the formula: % of Killing = 100 × (release of sample − spontaneous release)/(maximum release Triton X100 − spontaneous release)].

## Antibody-dependent-cell-cytotoxicity assay and MMoAK assay

NK cells were purified from fresh human PBMCs using EasySep NK cell enrichment kit (StemCell) according to the manufacturer's instructions and incubated overnight at 37 °C in complete medium. Purified NK cells were washed and incubated with eFluor670 labeled tumor cells at 5:1 E:T ratio, 37 °C, 5% $CO_2$. After 4h30 of incubation, cells were centrifugated and resuspended in FACS buffer containing a viability dye (Dapi or Sytox) and analyzed by flow cytometry. For MMoAK assay, autologous human PBMC were thawed and cultured at $1 \times 10^6$ cells/ml overnight at the incubator in complete medium. PBMCs were mixed with autologous Monocyte-Derived-Macrophages generated as previously described and incubated with eFluor670-labeled tumor cells in presence of 50% human serum (Sigma) for 48 h at 37 °C, 5% $CO_2$. In some experiments, either 2.8 ng/ml sCD38 (a concentration that is detected in MM patient's bone marrow[48]) or $0.5 \times 10^6$ RBCs were added to each well to assess the impact of CD38 and CD47 antigen-sink respectively. After completion of the assay, cells were centrifuged and resuspended in FACS buffer containing a viability dye (Dapi or Sytox). Flow-cytometric analysis was then performed with a Cytoflex (Beckman Coulter). Viable tumor cells were identified as positive for eFluor670 and negative for the viability cell stain. Absolute number of live tumor cells were then used to determine percentage of killing. For ADCC, killing was calculated with the following formula: % of Killing = 100 × (1-(Abs count/well of sample/Average of Abs count/well of no Ab)). For MMoAK: % of Killing = 100 × (1 - (Abs.count/well of sample/Average of Abs.count/well of Target only)).

## Coombs anti-IgG assay (Biorad®)

Extent of agglutination was evaluated using the Coombs anti-IgG assay (Biorad) according to manufacturer instructions. Briefly, a 0.8% RBCs suspension solution was prepared and loaded the columns of the kit. Increasing dose of antibodies were then added and incubated at 37 °C for 15 min before being centrifuged for 10 min at 1200 rpm. Extent of agglutination was scored from 0 (no agglutination) to 4 (complete agglutination).

## RBCs depletion assay

Red-blood cell depletion was evaluated in-house using a hematology analyzer. Briefly, blood was incubated for 1h30 at 37 °C under gentle agitation (100 rpm) in the presence of increasing dose of test and control antibodies. After incubation, cell counting was performed using Sigma 5H hematology analyzer (Swissavans) according to manufacturer's instructions. RBCs depletion was calculated as percentage of RBCs compared to baseline levels with the formula: % RBCs vs baseline= 100 − (1 − (RBCs concentration sample/RBCs concentration baseline)) × 100.

## Whole blood binding assay

A dose response of test and control antibodies was pre-incubated with AF647 or AF488-polyclonal goat Fab-anti-human IgG (Jackson Immunoresearch) in a 2:1 ratio for 15 min at room temperature. Pre-incubated antibodies were then added to fresh human peripheral blood and incubated for 30 min at 4 °C. For antibody binding to RBCs, whole blood was washed with FACS buffer and analyzed directly using Cytoflex flow cytometer (Beckman Coulter) by gating on RBCs population. Analysis was performed using Flowjo (BD).

## Hemolysis

Induction of hemolysis was tested in-vitro by incubating antibodies at 87 nM in fresh human peripheral blood at 37 °C. After 3h30 incubation, the absorbance of plasma (diluted 1/100) was read at 414 nm. Triton X-100 was used as a positive control for hemolysis. Results were normalized in GraphPad Prism software to no antibody treatment condition (negative control) and triton X-100 (positive control).

## ISB 1442- killing of patient-derived MM samples

Mononuclear (MNC) cells were isolated using Sepmate PBMC isolation tubes (Stemcell) and then incubated with increasing dose of antibodies in presence of 10% human serum (Sigma-Aldrich or Bern Transfusion Center) and 3 ng/ml human IL-6 (Peprotech), at 37 °C, 5% $CO_2$. After 20–24 h culture, cells were centrifugated and stained with the Live/Dead NIR viability dye (Thermofisher) for 30 min at 4 °C. After a washing step, cells were then stained with a mix of antibodies (Supplementary Table 5). After 30 min incubation with antibodies, cells were washed 2 times and finally resuspended in FACS Buffer. Live CD138$^+$ MM cells were quantified for each condition. A representative FACS gating is shown in Supplementary Fig. 11. The following formula was used to determine percentage of killing: % Killing= 100 x (1-absolute number cells treated condition/absolute number cells untreated condition). All antibodies were titrated with a dose-response from at least 1/50 with a serial dilution of 1/2 up to 1/3200 on positive cells and non-expressing cells. Choice of the optimal antibody dilution was based on the stain index calculation: Stain Index (SI) = (MFI of positive population − MFI of negative population)/(2 × SD of negative population).

## Cell-based competition assay with daratumumab

ISB 1442 and daratumumab were respectively labeled with AF488 and AF647 using SiteClick Antibody Azido Modification Kit and sDibo Alkyne kits (Thermofisher) according to manufacturer's instructions. Cells were then pre-incubated with saturating concentration of ISB 1442 purified antibody for 30 min at 4 °C before addition of saturating concentration of daratumumab-AF647 or ISB 1442-AF488 or a combination of daratumumab-AF647 and ISB 1442-AF488. Percentage of competition was calculated with the following formula: % competition =100 − (MFI sample × 100/MFI untreated).

## Live Imaging using Perkin Elmer Operetta

Live cell phagocytosis of tumor cells by macrophages was evaluated in-house using Perkin Elmer Operetta System. Cell-Trace Violet macrophages were incubated with CFSE or Far-Red labeled tumor cells in

presence of increasing dose of antibodies. Two fields per well were imaged every 5 minutes (26 time-points in total), for a period of 2 h with a 20× water immersion lens, confocal mode. Analyses of phagocytosis was performed by Perkin Elmer using a dedicated script.

## Platelet aggregation assays

This assay was performed by Platelet Services Ltd and performed according to their protocol and standards. Briefly, ISB 1442, daratumumab and hu5F9 was tested on Platelet-Rich-Plasma (PRP) prepared from citrated whole blood samples. The extent of platelet aggregation was measured by light transmission aggregometry (LTA) as an increase in light transmission through the sample using a specialized aggregometer, AggRAM (Helena Biosciences). LeoA1 antibody was used as a positive control for platelet aggregation.

## Statistics and reproducibility

Statistical analysis of in vitro experiments was conducted in GraphPad Prism software as indicated in the figure legends. No statistical test was used to determined sample size. Instead sample size was determined empirically according to previous knowledge of the variation in experimental setup. For potency testing in vitro, two exclusion criteria were used before EC50 extraction assessing the shape of the curve leading to the inclusion of regular pharmacological dose-dependent fit: $R^2$ superior to 0.7, to ensure the quality of the fit (regression used is sigmoidal dose response, GraphPad Prism software) and Span (difference between top fit and bottom fit) superior to 10% (window of response). With the exception of imaging experiments, only data that we were able to replicate at least in two independent experiments were included. For all in vivo experiments mice were randomized by tumor volume to first achieve the same average starting tumor volume then a similar standard deviation of tumor volumes between all groups. In the experiment reported in Fig. 4C–E outliers (by tumor volume) were excluded at randomization to better match standard deviations between groups. In Fig. 4A, B technicians were blinded to treatment groups. In Fig. 4C–E blinding was not possible and blinding was not conducted for Fig. 4F as treatment effects were not the outcome under evaluation. For in vitro experiments, the investigators were not blinded to allocation during experiments and outcome assessment.

## Reporting summary

Further information on research design is available in the Nature Portfolio Reporting Summary linked to this article.

# Data availability

Requests for new materials generated in this paper are to be directed to and will be fulfilled (pending MTA and associated restrictions) by the lead contacts. Source data are provided with this paper.

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

## Acknowledgements

We thank Emelie Svensson for performing the phage display panning for the discovery of anti-CD47 and anti-CD38 antibodies. We thank Mathilde Testut and Aurore Delachat for supporting the characterization of anti-CD47 and anti-CD38 antibodies. We thank Samitabh Chakraborti for supporting affinity maturation of anti-CD38 antibodies. We thank Amélie Laurendon, Jeremy Streuli, Olivier Bornert, Maureen Bardet, Dominique Schlicht Boyer, Caroline Craipeau and Matthew Blackburn for producing the antigens and supporting the generation and characterization of antibodies and antigens. We thank Fanny Broch, Aline Reynaud and Christelle Ries-Fécourt for the production and characterization of antibody constructs. We thank Perrine Suère and Blandine Pouleau for the screening and characterization of the initial candidates of the ISB 1442 project and for the setup of the MMoAK assay. We thank Emilie Nallet, Laura Carretero, Isabelle Gruber and Elodie Stainnack for helping with the processing of primary patients' material. We thank Marie-Agnès Doucey, Laure Bouchez, Rebecca Croasdale-Wood and Cian Stutz for their scientical contribution to the ISB 1442 project. We thank Cyrille Touzeau, Nicoletta Libera Lilli and Sophie Maïga for collecting and processing the patient samples from the MYRACLE (Myeloma Resistance and Clonal Evolution) cohort at CHU Nantes (France). We thank Cindy Lanvers for her help in collecting patient samples at University Hospital Geneva (Switzerland). We are grateful to Dr. Sarah Gooding and Mirian Salazar, all patients who donated samples and the HaemBio Biobank, MRC WIMM, Oxford, for provision of clinical samples. The authors acknowledge the Cytocell-Flow Cytometry and FACS core facility (SFR Bonamy, BioCore, Inserm UMS 016, CNRS UAR 3556, Nantes, France) for its technical expertise and help, member of the Scientific Interest Group (GIS) Biogenouest and the Labex IGO program supported by the French National Research Agency (no ANR-11-LABX-0016-01).

## Author contributions

G.C., E.C. and D.E. designed, performed most of the experiments and wrote the manuscript. M.T. and D.C. designed, performed experiments, and wrote the manuscript. L.J., F.J., C.L.N., C.M. designed and performed experiments. P.M., L.V., M.E., D.A.S. and R.A. performed experiments. D.A. designed in vivo studies and wrote the manuscript. M.T., K.Z., E.C.M., E.J.R., M.E., K.C., P.D.C., M.P. provided scientific and technical support as well as provided patients samples. B.S., M.M.L., S.A., D.M.R. and Z.E.A. provided scientific supervision and wrote the manuscript. P.M. and S.S. conceptualized the study and wrote the manuscript.

## Competing interests

D.E., P.M., M.T., L.J., L.V., D.A., D.C., D.A.S., C.L.N., D.M.R., Z.E.A., P.M. and S.S. are current employees of Ichnos Sciences. G.C., E.C., R.A., F.J., B.S., M.E., C.M., M.M.L. and S.A. are former employees of Ichnos Sciences. The remaining authors declare no competing interests.
