## [Peer Review File · Nature Communications]

Development of ISB 1442, a CD38 and CD47 bispecific biparatopic antibody innate cell modulator for the treatment of multiple myelomaREVIEWER COMMENTS

Reviewer #1 (Remarks to the Author):

This is very robust manuscript from Grandclément and collaborators describing the pre-clinical development of ISB 1442, a first-in-class CD38 and CD47 bispecific paratopic antibody for the treatment of myeloma (and potentially other CD38-expressing malignancies).

The manuscript is very well written, logic. The findings are properly documented with clear experiments. The work is very original and relevant given that this is a first-in-class antibody that is currently in clinical development. The authors clearly demonstrate the relevance of each element of the antibody through elegant experiments and addresses some possible pitfalls. I have only minor comments/suggestions to make.

In line 326 "Targeting CD38 with daratumumab has shown in MM patients a median overall survival (OS) of 50.1 months either as monotherapy or in combination" Is overstated and over simplified. Combination data comes from very different populations with very different impact of adding daratumumab. It is impossible to give "a number" to make a general statement about the OS impact of daratumumab. Consider informing that the drug has remarkable single agent activity and has improved survival (2 trials thus far) and PFS (multiple trials) when in combination.

Line 328 "relapses post-daratumumab therapy inevitably occur" is overstated. That is not proven in the newly diagnosed setting.

"daratumumab-refractory patients" in line 269. Disease is refractory, not patient. Consider "patients with daratumumab-refractory myeloma" or "killing of patient-derived daratumumab-refractory tumor cells".

"r/rMM" consider more customary abbreviation RRMM.

I missed seeing some brief discussions on limitations and potential challenges on the clinical development that could not be addressed with the clinical work. That would add balance to the manuscript.

Reviewer #2 (Remarks to the Author):

In this article, Grandclément et al. describe the engineering, development, and characterization of ISB 1442, an antibody targeting CD47 and two epitopes of CD38 to be used for treatment of multiple myeloma. The argument is made that biepitopic targeting of CD38 and monovalent targeting of CD47 allows for binding of both CD38-hi and -lo cells while avoiding on-target off-tumor toxicities associated with CD47 via low affinity and conditional binding. The Fc of the antibody is engineered for enhanced effector functions, where the potentiated ADCP and CDC functions are well characterized. The molecular structure presents some novel and therapeutically advantageous features, and the data are presented clearly and mostly support the conclusions. The article may benefit from additional experiments, namely comparison of ISB 1442 to an antibody that binds CD38 bivalently but monoepitopically (along with its existing CD47 binding) to further demonstrate the advantage of the biepitopic design.

Major points:

- To support the notion that biepitopic CD38 targeting is important for the activity of ISB 1442, it should be compared to an antibody that binds CD38 bivalently, but at the same epitope. The existing dummy controls certainly help to support this point, but they are not a direct comparison because they bind CD38 monovalently instead of bivalently. Thus, the reduction in activity of the dummy controls could be due to lower valency, in addition to monoepitopic targeting. Ideally, controls of ISB 1442 where both CD38 Fabs target the same epitope (e.g., a control where both CD38 Fabs contain B6-D9, and a control where both Fabs contain E2-RecA) should be constructed and tested as in Figure 2.

- For the competition figures (1A, S3A) the experiment should be performed in both "directions" (e.g., for figure S3A with both daratumumab and ISB 1442 used as the first antibody in separate experiments) to show that there is no unidirectional blocking. Additionally, a control should be included where the blocking antibody is not associated in the first step, to show the maximum binding of the second antibody in the absence of the first antibody. Otherwise, it is difficult to tell if the levels of binding seen in the second step are due to lack of saturation or truly due to lack of competition.
- How was figure 1B generated? I could not find the binning data cited on line 99, and also could not find any competition data for isatuximab. For this sort of binning data, I would expect to see a matrix of binding responses when each antibody is used as the blocking antibody or the second antibody. This data would strengthen the arguments made concerning competition.
- For figure S2 (comparing binding to Fc receptors), why was ISB 1442 compared to trastuzumab rather than a version of ISB 1442 with wild-type Fc (as was tested in Figure 2)? I understand the Fc portion of the antibody is the important part for this assay, but ISB 1442 has a different structure than a mAb like trastuzumab, and therefore this may not be the best comparator.

Minor points:

- Line 40 could be changed to "allowing for the generation of synthetic immunity..."
- On lines 98, 103, and 728 the phrase "in format" is ambiguous. Does this mean in the 2+1 bispecific format of ISB 1442?
- For figure 2F, it would be helpful to explain why CD47_DU has such high maximal killing, although the potency is weaker. If the monovalent CD47 binding is truly dependent on engagement of CD38, wouldn't the expectation be to see no killing from CD47_DU?
- On line 157 it is stated that the Fc-optimized domain "enhances all Fc effector mechanisms", but in this section only ADCP and CDC are tested (no ADCC data is shown). I would suggest either rephrasing this sentence (e.g., "enhances ADCP and CDC activities"), or including ADCC data in the supplementary figures.
- For the description of supplementary figure 4 (line 165), one could speculate that the higher binding maximum of ISB 1442 is due to biepitopic binding. If one molecule of ISB 1442 is able to bind simultaneously via both its CD38-targeting arms to one molecule of CD38 receptor (as opposed to one molecule of daratumumab, which could bind to two molecules of CD38 receptor), one might expect higher maximal binding for ISB 1442 as a result of daratumumab saturating the surface receptors at half the amount of antibody bound. This could alternatively be discussed in the paragraph beginning on line 291.
- On line 166, it should be noted that while ISB 1442 showed comparable maximal phagocytosis, it had a significantly higher EC50.
- For the legend of Figure 4A, I would recommend changing the language to "daratumumab at 16mg/kg twice weekly" to avoid the ambiguous word "biweekly", which could also mean every two weeks.
- There is no figure legend for Figure 4E-F.
- On line 267, the citation should be for Figure 6E, not Figure E.
- On line 281, the sentence should begin "The flexibility..."
- For the methods section describing the BLI competition assays and SIRPa blocking assays, the concentrations of each protein should be specified.
- For figure 1F, the CH1 domain is not shown in the legend. Also, to clarify, is the same light chain used for all three Fabs?
- Some of the figures are a bit blurry [e.g., Figure 2, Figure 3 (especially panel G), Figure 4 (especially panel F), Figure 5, Figure S7]. Are higher resolution versions available?
- On line 753, it could read, "...induced by ISB 1442 or its dummy controls."
- For figure 4D, it's hard to make out the line for ISB 1442 (s.c.). Is it under the orange line? Could this be clarified somehow?
- In the legend for Figure S2 and S6, it could be changed to "monocyte-derived macrophages."
- For figure S8, how does the concentration of soluble CD38 used in the experiment compare to the amount found in MM patients?

Point-by-point response to the reviewers' comments

Reviewer #1 (Remarks to the Author):

- 1) *In line 326 "Targeting CD38 with daratumumab has shown in MM patients a median overall survival (OS) of 50.1 months either as monotherapy or in combination" Is overstated and over simplified. Combination data comes from very different populations with very different impact of adding daratumumab. It is impossible to give "a number" to make a general statement about the OS impact of daratumumab. Consider informing that the drug has remarkable single agent activity and has improved survival (2 trials thus far) and PFS (multiple trials) when in combination.*

We thank the reviewer for the comment. We have now changed the sentence to:

"Targeting CD38 with daratumumab has shown remarkable activity in MM patients improving their survival both as single agent or in combination with SOC "

- 2) *Line 328 "relapses post-daratumumab therapy inevitably occur" is overstated. That is not proven in the newly diagnosed setting.*

We have now changed the sentence to: "Since relapses post-daratumumab therapy can occur."

- 3) *"daratumumab-refractory patients" in line 269. Disease is refractory, not patient. Consider "patients with daratumumab-refractory myeloma" or "killing of patient-derived daratumumab-refractory tumor cells".*

We thank the reviewer for the comment. We have now changed the sentence to:

"patients with daratumumab-refractory MM."

- 4) *"r/rMM" consider more customary abbreviation RRMM.*

We have changed the acronym to RRMM throughout the manuscript.

- 5) *I missed seeing some brief discussions on limitations and potential challenges on the clinical development that could not be addressed with the clinical work. That would add balance to the manuscript.*

We thank the reviewer for pointing out this omission. We have now added a section in the discussion describing the limitations and open questions related to ISB 1442 development in the clinic.

Reviewer #2 (Remarks to the Author):

Major:

- 1) *To support the notion that biepitopic CD38 targeting is important for the activity of ISB 1442, it should be compared to an antibody that binds CD38 bivalently, but at the same epitope. The existing dummy controls certainly help to support this point, but they are not a direct comparison because they bind CD38 monovalently instead of bivalently. Thus, the reduction in activity of the dummy controls could be due to lower valency, in addition to monoepitopic targeting. Ideally, controls of ISB 1442 where both CD38 Fabs target the same epitope (e.g., a control where both CD38 Fabs contain B6-D9, and a control where both Fabs contain E2-RecA) should be constructed and tested as in Figure 2.*

We thank the reviewer for the comment. We have generated monoparatopic bivalent variants of ISB 1442 that bind bivalently to the same CD38 epitope in a 2+1 format and compared them to ISB 1442 (2+1 biparatopic antibody). We believe that these data answer the specific point addressed by the reviewer and that they reinforce our conclusions regarding the benefit of biparatopic targeting; we have generated a new figure (Figure 3) and appropriately modified the Results and Discussion sections.

- 2) *For the competition figures (1A, S3A) the experiment should be performed in both "directions" (e.g., for figure S3A with both daratumumab and ISB 1442 used as the first antibody in separate experiments) to show that there is no unidirectional blocking. Additionally, a control should be included where the blocking antibody is not associated in the first step, to show the maximum binding of the second antibody in the absence of the first antibody. Otherwise, it is difficult to tell if the levels of binding seen in the second step are due to lack of saturation or truly due to lack of competition.*

We thank the reviewer for the thoughtful comment. We have now performed the competition assay employing both orientations and added a control, in which the blocking antibody is not associated in the first step. These new data are included in Figure 1C and Supplementary Figure 3A (S3A). In brief, the competition assay employing antibodies in "both orientations" and including the control without antibody in the saturating phase were repeated. Figure 1 and S3A were edited appropriately and now include a binning matrix with binding responses.

- 3) *How was figure 1B generated? I could not find the binning data cited on line 99, and also could not find any competition data for isatuximab. For this sort of binning data, I would expect to see a matrix of binding responses when each antibody is used as the blocking antibody or the second antibody. This data would strengthen the arguments made concerning competition.*

A similar approach to the one described above was used to assess competition with isatuximab. These data have been included in a new Figure (Supplementary Figure 2) and in panel B of Figure 1. Figure 1B was also edited by including enlarged epitope bin ellipses for a more accurate interpretation of the binning data, and it includes daratumumab and isatuximab known epitopes.

- 4) *For figure S2 (comparing binding to Fc receptors), why was ISB 1442 compared to trastuzumab rather than a version of ISB 1442 with wild-type Fc (as was tested in Figure 2)? I understand the Fc portion of the antibody is the important part for this assay, but ISB 1442 has a different structure than a mAb like trastuzumab, and therefore this may not be the best comparator.*

Fc receptor affinity measurements of ISB 1442 with a IgG1 (WT-Fc) or silenced Fc were performed. SPR values are now included in Supplementary Table 2.

Minor:

- 5) *Line 40 could be changed to “allowing for the generation of synthetic immunity...”*

The text was modified according to the reviewer’s suggestion.

- 6) *On lines 98, 103, and 728 the phrase “in format” is ambiguous. Does this mean in the 2+1 bispecific format of ISB 1442?*

The text was modified to indicate the 2+1 bispecific format.

- 7) *For figure 2F, it would be helpful to explain why CD47_DU has such high maximal killing, although the potency is weaker. If the monovalent CD47 binding is truly dependent on engagement of CD38, wouldn’t the expectation be to see no killing from CD47_DU?*

We thank the reviewer for the comment. For this specific experiment we used a cell line (KMS-12-BM) that expresses low levels of CD38. This cell line expresses very high levels of CD47 (around 150,000 molecules per cells). We did not detect any binding of the CD47-DU molecule on these cells (Figure below), however, as mentioned by the reviewer, we detected some induction of phagocytosis at high concentration. Our interpretation of this phenomenon is as following: at high concentration the molecule can bind to macrophages via its enhanced Fc domain (especially in the absence of competition for the Fc Receptor in vitro) and due to the high expression level of CD47 in this cell line, induce transient blocking of some of the CD47 sufficient to initiate phagocytosis.

- 8) *On line 157 it is stated that the Fc-optimized domain “enhances all Fc effector mechanisms”, but in this section only ADCP and CDC are tested (no ADCC data is shown). I would suggest either re-phrasing this sentence (e.g., “enhances ADCP and CDC activities”), or including ADCC data in the supplementary figures.*

We thank the reviewer for the comment. The text has been amended to reflect the fact that the Fc optimization enhances ADCP and CDC.

- 9) *For the description of supplementary figure 4 (line 165), one could speculate that the higher binding maximum of ISB 1442 is due to biepitopic binding. If one molecule of ISB 1442 is able to bind simultaneously via both its CD38-targeting arms to one molecule of CD38 receptor (as opposed to one molecule of daratumumab, which could bind to two molecules of CD38 receptor), one might expect higher maximal binding for ISB 1442 as a result of daratumumab saturating the surface receptors at half the amount of antibody bound. This could alternatively be discussed in the paragraph beginning on line 291.*

We thank the reviewer for the comment. We have now added a new sentence: “This is consistent with the biparatopic nature of ISB 1442 as opposed to that of monoparatopic daratumumab, which saturates the CD38 surface receptors at half the amount of ISB 1442 antibody “

10) *On line 166, it should be noted that while ISB 1442 showed comparable maximal phagocytosis, it had a significantly higher EC50.*

We changed the text to reflect that the lower EC50 is induced by daratumumab in the CD38-high expressing cell line.

11) *For the legend of Figure 4A, I would recommend changing the language to “daratumumab at 16mg/kg twice weekly” to avoid the ambiguous word “biweekly”, which could also mean every two weeks.*

We amended the text according to the reviewer’s comment.

12) *There is no figure legend for Figure 4E-F.*

The missing legend has been added.

13) *On line 267, the citation should be for Figure 6E, not Figure E.*

The missing number is added in the text (now Figure 7 E)

14) *On line 281, the sentence should begin “The flexibility...”*

We amended the text according to the reviewer’s comment.

15) *For the methods section describing the BLI competition assays and SIRPa blocking assays, the concentrations of each protein should be specified.*

The concentrations of each protein are now specified.

16) *For figure 1F, the CH1 domain is not shown in the legend. Also, to clarify, is the same light chain used for all three Fabs?*

CH1 domain has been added to the figure 1F and the legend now reads: “All Fab domains make use of an identical common light chain (cLC)”.

17) *Some of the figures are a bit blurry [e.g., Figure 2, Figure 3 (especially panel G), Figure 4 (especially panel F), Figure 5, Figure S7]. Are higher resolution versions available?*

We thank the reviewer for this observation. Figures with higher resolution will be used in the final format of the manuscript.

18) *On line 753, it could read, "...induced by ISB 1442 or its dummy controls."*

In the figure legend, a new sentence was added: "F-M. ADCP and CDC induced by ISB 1442 or its dummy controls."

19) *For figure 4D, it's hard to make out the line for ISB 1442 (s.c.). Is it under the orange line? Could this be clarified somehow?*

We thank the reviewer for the comment. In the new version of the Figure 4D it is now visible that the two lines overlap.

20) *In the legend for Figure S2 and S6, it could be changed to "monocyte-derived macrophages."*

We have changed the text to: "monocyte-derived macrophages "

21) *For figure S8, how does the concentration of soluble CD38 used in the experiment compare to the amount found in MM patients?*

We thank the reviewer for the comment. The concentration used in the experiment compares to the sCD38 found in the bone marrow of MM patients. This information is now added in the relevant Materials and Methods section along with the reference to the publication where this was published: "Li, T. et al. Nanobody-based dual epitopes protein identification (DepID) assay for measuring soluble CD38 in plasma of multiple myeloma patients. *Anal Chim Acta* 1029, 65-71, doi:10.1016/j.aca.2018.04.061 (2018)."

REVIEWERS' COMMENTS

Reviewer #1 (Remarks to the Author):

I believe the authors properly addressed all the reviewer's concerns.

Reviewer #2 (Remarks to the Author):

The authors have addressed all my concerns. I have no further comments.